# Visualizing single-molecule conformational transition and binding dynamics of intrinsically disordered proteins

Wenzhe Liu[1,6], Limin Chen[2,6], Dongbao Yin[1], Zhiheng Yang[1], Jianfei Feng ®[1], Qi Sun ®[1] ✉, Luhua Lai ®[1,2,3] ✉ & Xuefeng Guo ®[1,4,5] ✉

Intrinsically disordered proteins (IDPs) play crucial roles in cellular processes and hold promise as drug targets. However, the dynamic nature of IDPs remains poorly understood. Here, we construct a single-molecule electrical nanocircuit based on silicon nanowire field-effect transistors (SiNW-FETs) and functionalize it with an individual disordered c-Myc bHLH-LZ domain to enable label-free, in situ, and long-term measurements at the single-molecule level. We use the device to study c-Myc interaction with Max and/or small molecule inhibitors. We observe the self-folding/unfolding process of c-Myc and reveal its interaction mechanism with Max and inhibitors through ultrasensitive real-time monitoring. We capture a relatively stable encounter intermediate ensemble of c-Myc during its transition from the unbound state to the fully folded state. The c-Myc/Max and c-Myc/inhibitor dissociation constants derived are consistent with other ensemble experiments. These proof-of-concept results provide an understanding of the IDP-binding/folding mechanism and represent a promising nanotechnology for IDP conformation/interaction studies and drug discovery.

Intrinsically disordered proteins (IDPs) or regions (IDRs), which were discovered in the 1990s, lack stable secondary and tertiary structures in physiological conditions[1,2]. These unfolded and flexible proteins are abundant in nature and exist as dynamic conformational ensembles[3]. They play essential roles in various critical biological processes, including regulation of transcription and translation[4], post-translational modification[5,6], cell signaling[7], and molecular recognition[8,9]. As a result of their multiple biofunctions, IDPs are frequently implicated in human diseases, including cancer, viral infection, and developmental disorders[10,11], and they are regarded as potential novel drug targets[12,13].

Recently, several important functional IDP systems, such as p53-MDM2[14,15], EWS-FLI1[16,17], and c-Myc-Max[18–24], have been studied

extensively. The proto-oncogenic c-Myc is an essential regulator in multiple biological processes, such as cell cycle and cellular metabolism. With a *C*-terminal basic helix-loop-helix zipper (bHLH-Zip) domain, the disordered c-Myc can specifically interact with its partner protein Max, forming a heterodimeric structure that regulates gene expression[18,24,25]. Overexpression of c-Myc causes disorder in cell proliferation and signaling pathways[23], resulting in a wide range of human cancers[25]. Therefore, inhibition of its function has become a potential therapy for cancer, despite the challenges of directly targeting the functional disordered region of IDPs[26,27].

The unstructured nature of IDPs presents great challenges for drug discovery; however, their molecular recognition features provide

[1]Beijing National Laboratory for Molecular Sciences, College of Chemistry and Molecular Engineering, Peking University, 292 Chengfu Road, Haidian District, 100871 Beijing, P. R. China. [2]Peking-Tsinghua Center for Life Sciences, Peking University, 100871 Beijing, P. R. China. [3]Center for Quantitative Biology, Academy for Advanced Interdisciplinary Studies, Peking University, 100871 Beijing, P. R. China. [4]Center of Single-Molecule Sciences, Institute of Modern Optics, Frontiers Science Center for New Organic Matter, College of Electronic Information and Optical Engineering, Nankai University, 38 Tongyan Road, Jinnan District, 300350 Tianjin, P. R. China. [5]National Biomedical Imaging Center, Peking University, Beijing 100871, P. R. China. [6]These authors contributed equally: Wenzhe Liu, Limin Chen. ✉e-mail: qsun2015@pku.edu.cn; lhlai@pku.edu.cn; guoxf@pku.edu.cn

useful information for designing molecules that bind with them[28]. A number of compounds that target c-Myc have been reported[18,24,25,29–31], such as 10074-A4, which was discovered by high-throughput screening, and PKUMDL-YC-1205, which was found through a "multi-conformational-affinity" computational strategy[32]. These compounds were shown to directly bind the disordered bHLH-Zip domain of c-Myc. Circular dichroism (CD) and nuclear magnetic resonance (NMR) experiments, as well as molecular dynamics simulations, indicated that c-Myc undergoes conformational changes upon binding to inhibitors, thus interrupting its heterodimerization with Max[24,32–34]. However, the underlying dynamic nature and detailed interaction mechanisms of c-Myc still need urgent exploration.

CD, fluorescence, and NMR spectroscopies provide information about the conformational ensemble of IDPs in their apo states, as well as about their dynamic conformational changes upon binding with partner molecules[35]. These ensemble experiments have shown that the free state of the *C*-terminus of c-Myc contains a partial helical structure[20] and have provided fundamental thermodynamic and kinetic characteristics of Myc-Max interactions[36]. However, direct detection of the transient folding process of c-Myc and its interaction with Max is challenging. Although several models have been proposed to explain the interactions between IDPs and their partners, such as coupled folding[12], binding-induced-folding[37], and conformational selection[38], steady intermediate states of c-Myc to undergo conformational transition have not been distinguishable by conventional methods[36]. Single-molecule experimental methods, such as single-molecule fluorescence[39–42], atomic force microscopy[43–45], and nanopore techniques[46,47], have been applied to provide insights into structural and dynamic heterogeneity in the folding and binding processes of IDPs. Nevertheless, long-timescale assessment of single-molecule IDPs with high temporal resolution is still a formidable challenge.

Here, we build a c-Myc-modified molecular nanocircuit on a single-molecule electrical device platform with silicon nanowire field-effect transistors (SiNW-FETs) as biosensors (Fig. 1a), which is capable of label-free, real-time biodetection at the single-molecule and single-event level[48–52]. We focus on the *C*-terminal fragment of c-Myc (c-Myc$_{370–414}$ with a cysteine at the *C*-terminal, hereafter, "LC46") and its interaction with Max (Max$_{37–83}$, hereafter, "DQ47"). We directly observe the self-folding process of LC46 and its interaction with DQ47 as well as with small molecule inhibitors (10074-A4 and PKUMDL-YC-1205). We recognize different encounter intermediates of LC46

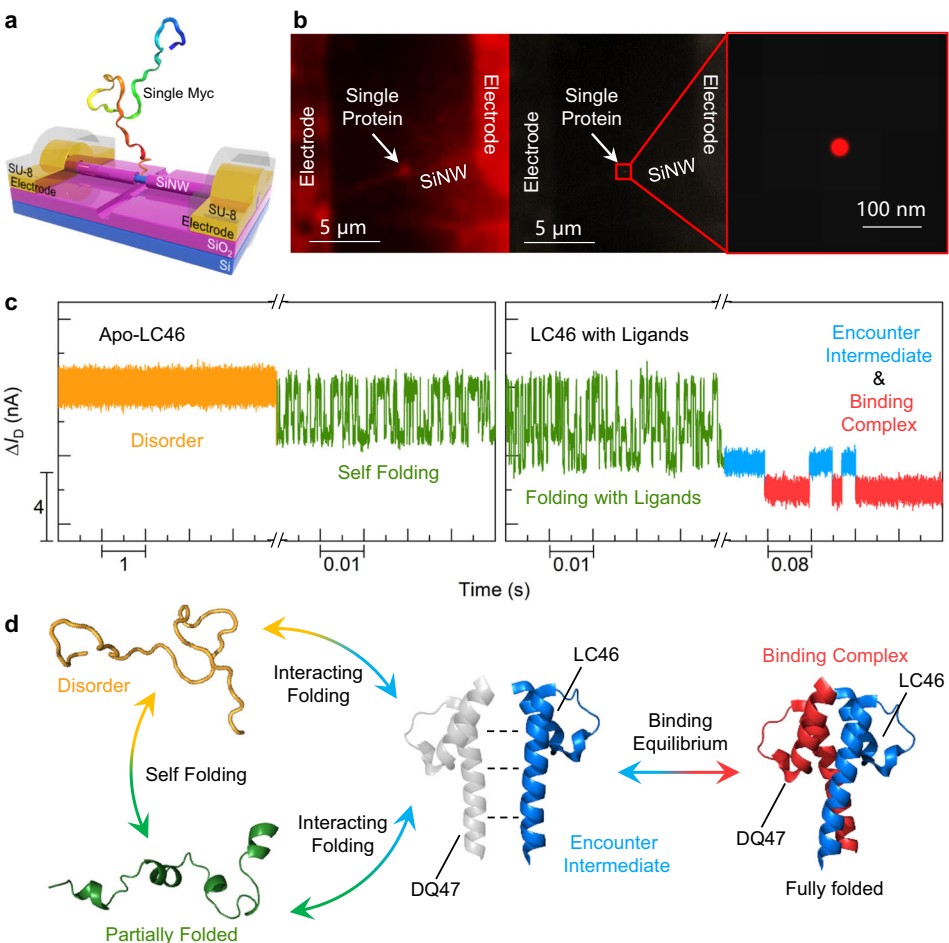

**Fig. 1 | Schematic demonstration and characterization of a c-Myc-modified SiNW-FET single-molecule device. a** Schematic diagram of the device architecture (blue for Si layer, magenta for SiO$_2$ layer, gold for Au electrodes, and light gray for SU-8 layer). **b** Stochastic optical reconstruction microscopy characterization of c-Myc single-molecule modification. The left panel shows the single-LC46-modified device in the dark field, taken under the excitation laser of 480 nm wavelength. The medium panel is the stochastic optical reconstruction of 5000 photos. The right panel is the magnification of the single reconstruction spot (red) in the medium column. **c** Real-time current trajectories (time resolution of 17 μs) at different stages from left to right: unstructured (disordered) state of Myc (yellow), the transient self-folding process (green in the left panel), the Myc folding process with Max or ligands (green in the right panel), and Myc-Max (ligands) binding equilibrium (blue for encounter intermediate and red for binding complex). **d** Schematic diagram of c-Myc conformations at different interaction stages (yellow for disorder state, green for partially folded state, gray for unbound/interacting DQ47, blue for LC46, and red for binding DQ47). The heterodimer structure was adapted from PDB "1NKP [https://doi.org/10.2210/pdb1NKP/pdb]". Source data are provided as a Source Data file.

through competitive binding between inhibitors and DQ47, which further verify the dynamic interaction pathway.

## Results

### Device fabrication and real-time electrical monitoring

The SiNW-FET-based single-molecule device was fabricated through point modification and a confinement reaction on the side wall of a SiNW[48,53,54]. Briefly, after the fabrication of the SiNW-FET array, a gap-opening process was performed using electron beam lithography to create a nanogap on the side wall of the SiNW and expose the Si-H surface after wet-etching by HF-NH$_4$F buffer. Then, through alkyne hydrosilylation of Si-H bonds with undecynoic acid and N-hydroxysuccinimide esterification, active ester terminals were formed for subsequent conjunction of N-(2-aminoethyl) maleimide. Finally, LC46, with its terminal cysteine residue, was connected to the surface of the SiNW by a Michael addition reaction between sulfhydryl and maleimide groups (Fig. 1a). The materials and detailed methods are provided in the Supplementary Information (Supplementary Notes 1 and 2, and Supplementary Figs. 1–16).

Labeling reaction with fluorescein isothiocyanate isomer I (FITC isomer I) was used to characterize the single-LC46 functionalization. FITC isomer I reacted with the lysine amino group in the LC46 sequence to realize the covalent addition of FITC and provide a luminescent group for subsequent fluorescence characterization using stochastic optical reconstruction microscopy. Fluorescence images and stochastic reconstruction results showed a single location spot with a resolution of ~20 nm, corresponding to a single fluorescence molecule (Fig. 1b and Supplementary Fig. 17). The device without FITC modification was also characterized using atomic force microscopy (Supplementary Fig. 18), verifying that the device was successfully modified with a single LC46 molecule. The single-LC46-modified devices were used for single-molecule sensing. In all real-time electrical (DC) measurements, the source-drain and gate voltages were set to 300 mV and 0 mV by a lock-in amplifier, respectively (Supplementary Note 3). The source-drain currents passing through the SiNW-device were amplified by a preamplifier and collected by a low-pass filter with a bandwidth of 10 kHz and a sampling rate of 57.6 or 28.8 kHz (17.4 or 34.7 μs), which presented various signal patterns corresponding to different behaviors of LC46 (Fig. 1c, d).

In control experiments, bare and maleimide-modified devices were monitored in a blank condition (phosphate-buffered saline [PBS, ×0.01, 8 g/L NaCl, 0.2 g/L KCl, 1.44 g/L Na$_2$HPO$_4$ and 0.24 g/L KH$_2$PO$_4$], 5% dimethylsulfoxide [DMSO], pH = 7.4). There were no obvious current signal fluctuations during the measurements, only the 1/$f$ noise from the device itself and the detection environment (Supplementary Fig. 19a, b), indicating that there was no obvious interaction between these devices and the buffer.

In the measurement of Myc-modified devices, the current trajectories showed a long-last single current stage with transient clustered signals (in 0.01× PBS, 5% DMSO, pH = 7.4) (Figs. 1c and 2a, and Supplementary Fig. 19c). A 0.01× PBS solution was used to acquire the appropriate Debye length and enough signal-to-noise ratio according to the PBS concentration-dependent experiments (Supplementary Figs. 20 and 21, Supplementary Table 1). Kinetic analysis results from surface plasmon resonance (SPR) experiments indicated that the binding activity of Myc and Max remained under this low salt concentration (Supplementary Table 2). The prolonged single current state observed was not considered to be a specific conformation but rather a conformational ensemble encompassing multiple conformations. In the period of the single current state, the LC46 peptide resided within an energy landscape without prominent energy wells. The transitions between different conformations occurred at a rapid pace, on the timescale ranging from sub-μs to μs. These conformational changes were beyond the limitation of the sampling rate and difficult to capture. The cluster oscillatory signals were considered to arise

when the peptide briefly occupies the deeper potential wells on the energy landscape, leading to the formation of distinct conformational ensembles characterized by longer dwell times, which can be captured. These clustered signals showed current decreases and could mainly be divided into three states, each with a short dwell time (at microsecond scale). According to the charge transport mechanism of p-type SiNWs, the enrichment of surface positive charges will lead to a decrease in conductance[48,50,52,55]. LC46 (pI = 8.94) used in these experiments was positively charged (+2e) in PBS at pH = 7.4. By inference, the clustered current decrease originates from the transient coiled and partially folded state of LC46. The coiled ensembles of the IDR (i.e., transient partial folding of the IDR) lead to a relatively dense spatial distribution of amino acid residues, resulting in a transient enrichment of total positive charges near the point defects of the SiNWs and thus a transient current decrease (Supplementary Fig. 22). According to the time resolution of 17 μs in the experiments, the instantaneous signal rise and decrease between the current states corresponded to the changes between conformational ensembles rather than conformational changes (~1–10 μs)[56].

The fast-current pulses within the clustered signals indicated a three-state population (Fig. 2a), which corresponds to the rapid transition between different conformational ensembles of LC46 that have different distributions of secondary structures. These transient clustered current decreases were negligible in comparison with the dominant disordered state (highest current level) if only average analysis is carried out. However, when LC46 folds into a certain conformational ensemble through random thermal motion, it will fall into the potential well in a partially structured state and the transition from disorder to the partially helical state can be captured, which results in the transient clustered current signals in the SiNW-FET-based single-molecule device. Conformations within the same ensemble tend to have similar degrees of folding, thereby presenting convergent kinetic and thermodynamic properties. The short duration of these clustered signals in our experiments was in accordance with the predominantly disordered nature of LC46[20,32,33].

We then performed a series of temperature-dependent experiments to analyze the clustered current signals generated by transient folding of LC46 (Fig. 2b and Supplementary Fig. 23). Within the clustered current signals, three conformational ensembles were observed at all temperatures but with different distributions. The overall duration of the clustered folding signal was up to the second scale, and the dwell time of single pulses (for one certain conformational ensemble) was on the hundred-microsecond scale.

QuB software was used to analyze these current signals[57]. The dwell times of the three current states were extracted based on a hidden Markov model, and the statistical dwell time distributions of different current states were fitted (Fig. 2c, Supplementary Fig. 24 and Supplementary Table 3; 0, 1, and 2 represent high, medium, and low current states, respectively). Different populations (state 1 as well as state $1^*_{LT}$ at lower temperature or state $1^*_{HT}$ at higher temperature) were found in the dwell time distributions of the medium current state, indicating that at the same temperature, there were two conformational ensembles with similar effective gate voltage but different kinetic behaviors. The dynamic behavior of state $1^*_{LT}$ and state $1^*_{HT}$ showed a distinct temperature-dependent nature. Interestingly, state $1^*_{LT}$ only existed at lower temperatures (25, 30 °C), and $1^*_{HT}$ emerged at higher temperature (45 °C), which were barely captured near physiological temperature (35–40 °C) (Fig. 2c, d). These conformational ensemble intermediates (Fig. 2e) demonstrated a dynamic self-folding process of the free state c-Myc.

### Myc-Max interaction

We next studied the Myc-Max interaction process using different concentrations of DQ47 (i.e., the region of Max that interacts with Myc) under pseudo-physiological conditions (pH = 7.4, 37 °C). In the low

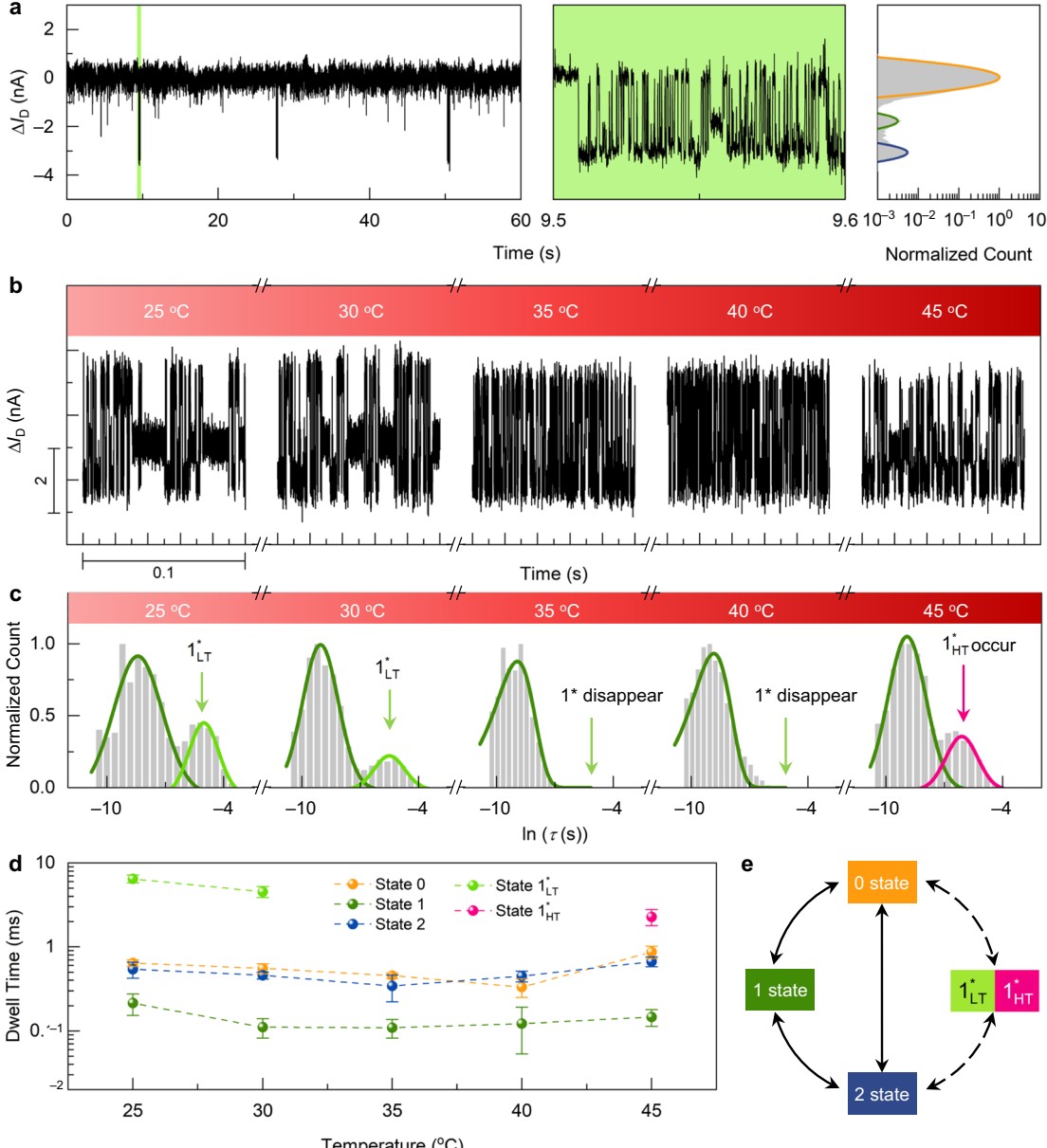

**Fig. 2 | Real-time monitoring of the c-Myc self-folding process. a** Real-time current trajectory (time resolution of 17 µs) of the c-Myc-modified device in a blank buffer solution at 25 °C (phosphate-buffered saline [PBS, ×0.01, 8 g/L NaCl, 0.2 g/L KCl, 1.44 g/L Na$_2$HPO$_4$ and 0.24 g/L KH$_2$PO$_4$], 5% dimethylsulfoxide [DMSO], pH = 7.4). The left panel shows a 60-s real-time trajectory, the middle panel shows a magnification of the clustered current signals (green part in the left panel), and the right panel is a histogram of the current data (yellow, green and blue for 0, 1 and 2 current states). **b** Current pulses within the clustered signals at different temperatures. **c** Dwell time distribution of the medium current states (States 1 [dark green], 1$^*_{LT}$ [light green] and 1$^*_{HT}$ [magenta]) at different temperatures. **d** Dwell time of different current states corresponding to different conformational intermediates ($n$ = 3, number of the devices, data are presented as mean values ± SD). **e** Transition relationships between different conformational intermediate ensembles. Dotted lines indicate that the process may disappear at certain temperatures. Source data are provided as a Source Data file.

DQ47 concentration range (5–500 nM), sophisticated multi-state current signals were observed during real-time monitoring (0.01× PBS with DQ47, 5% DMSO, pH = 7.4) (Fig. 3a, Supplementary Fig. 25a–e). Control experiments showed no obvious signals when a device without LC46 was measured in DQ47 solution, indicating that signals in Fig. 3a originated from the interaction between LC46 and DQ47 (Supplementary Fig. 26a). These current signals mainly exhibited two different patterns (Fig. 3b). One was a clustered pulse signal pattern with multiple current states on a microsecond time scale, and the other was a bistate signal pattern on a millisecond time scale. These two signal patterns alternated with each other during long measurements. With the increase of DQ47 concentration, the duration of the fast-clustered

signals gradually reduced, and when the Max concentration reached over 1 µM, the slow bistate signals became predominant (Fig. 3c and Supplementary Fig. 25f, g). The high-frequency clustered signal presented a similar kinetic behavior (pulse signals with dwell time ~100 µs) to the clustered signals observed in apo-LC46 experiments. The uniform and regular bistate signals at high DQ47 concentrations occurred with a lower conductance.

According to the chronological order and the alternating nature of the current signals, we assumed that the high-frequency clustered current signals represented the folding process of LC46 with multiple labile intermediate conformation ensembles. The bistate signals with longer dwell times originated from the dimerization process of LC46

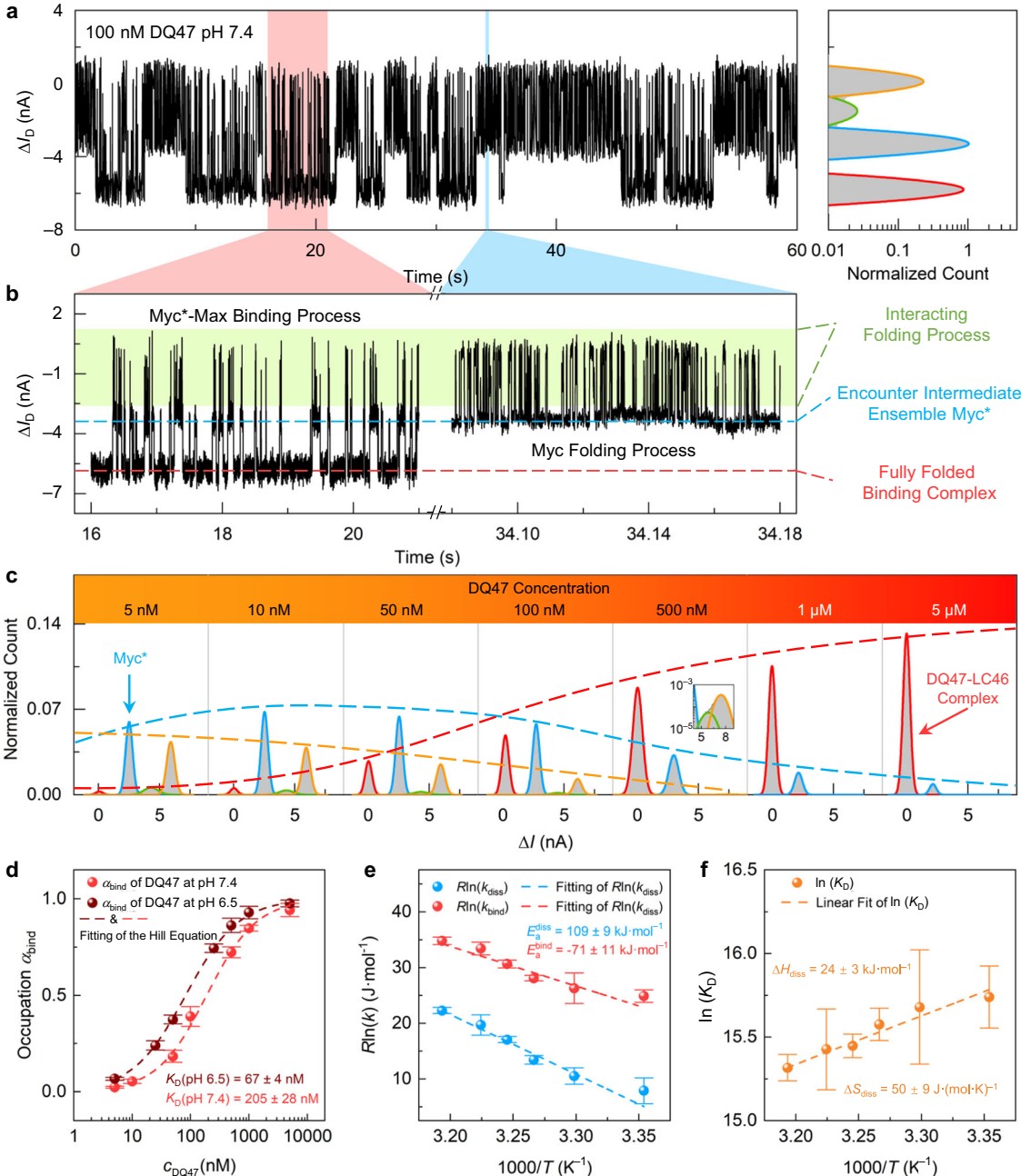

**Fig. 3 | Real-time monitoring of the Myc-Max interaction. a** Real-time trajectory over 60 s of the Myc-Max interaction process and a histogram of the current data (time resolution of 17 µs). **b** Magnification of current signal patterns; Myc folding process from the blue marked area in (**a**), and the Myc*-Max binding process from the red marked area in (**a**). The horizontal marks show the different stages during the interaction: green for the interacting folding process; blue for encounter intermediate ensemble, Myc*; and red for the fully folded binding complex. **c** Histogram of current data as a function of DQ47 concentration (DQ47 is the region of Max that interacts with Myc). Red indicates Myc*, blue indicates the binding state, and yellow and green indicate partial folding states. **d** Population of the Myc-Max bound state at different DQ47 concentrations and different pHs (red

for pH 7.4 and dark red for pH 6.5, $n = 3$, number of the devices, data are presented as mean values ± SD). Dissociation constants were derived from the Hill equation (Supplementary Table 4). **e** The rate constant $k$ at different temperatures (Arrhenius diagram, red and blue for binding and dissociation process). The activation energy of binding and dissociation were obtained by fitting the Arrhenius equation ($n = 3$, number of the devices, data are presented as mean values ± SD). **f** Equilibrium constants of the Myc*-Max binding process at different temperatures. The enthalpy change and entropy change of the dissociation process were obtained by linear fitting ($n = 3$, number of the devices, data are presented as mean values ± SD). Source data are provided as a Source Data file.

and DQ47. The lower current state in the bistate signals corresponded to the fully folded binding complex, which was much more stable than the partially folded conformational ensembles. Because of the global positive charge of DQ47 (+2e at pH 7.4) and LC46, as well as the compact HLH-zip structure, a significant change in the effective gating voltage occurred, leading to a lower current level (Fig. 3b, c). The gating voltage was mainly affected by charge distribution changes

induced by the dramatic conformational changes. Interestingly, the signals always went through a certain current state to finally form the binding complex, which was referred to as an encounter intermediate ensemble that bridges the labile conformation ensemble of free c-Myc to the fully folded state in the binding complex. Based on the concentration-dependent signal patterns, we propose an interaction hypothesis (Fig. 1d) for LC46 and DQ47. At a very low DQ47

concentration (5 nM), LC46 may first undergo a loose interaction system and fold to a dominant Myc*, which is favorable for further specific binding. This encounter intermediate ensemble was further enlarged along with the increase of DQ47 concentration, according to the increase of Myc* population (5–100 nM DQ47 in Fig. 3c). As DQ47 concentration further increased, the equilibrium of the binding process would finally shift to the fully folded binding structure (100 nM−5 µM DQ47 in Fig. 3c). DQ47 concentration-dependent experiments at pH 6.5 demonstrated the same interaction mechanism (Supplementary Fig. 27). As the arithmetic averaged dwell times were calculated second by second, a dynamic disorder analysis was performed to explore the fluctuation of the conformational changes in the binding equilibrium[58]. The arithmetic averaged dwell times of Myc* exhibited a distribution with a relatively wide range, implying a larger conformational fluctuation of the encounter intermediate ensemble in comparison with the fully folded complex (Supplementary Fig. 28).

Temperature-dependent experiments were carried out to investigate the thermodynamic properties of Myc-Max binding. At a high DQ47 concentration (1 µM), the current trajectories exhibited sustained bistate signals, indicating that the binding and dissociation process of Myc* and Max was predominant (Supplementary Fig. 29). According to the previous discussion, the lower current state was the fully bound state in which LC46 dimerized with the positively charged DQ47, while the higher current state was the encounter conformational intermediate Myc* formed by the transient release of DQ47 from the heterodimer. As the temperature increased, the frequency of transition between bound and dissociated states gradually increased while the corresponding current distributions remained unchanged. When the temperature was increased to 45 °C, higher current states emerged, indicating that Myc* became unstable at this temperature; that is, conformational intermediates that were unfavorable for binding began to appear, thus affecting the Myc-Max binding affinity.

From DQ47 dose-response experiments, the population tendency of Myc-Max binding ($\alpha_{bind}$) was illustrated as a function of DQ47 concentration (Fig. 3d). The dissociation constant $K_D$ of the Myc-Max complex at different pHs was obtained by fitting the Hill equation (Fig. 3d and Supplementary Table 4)[32]. The $K_D$ of DQ47 binding to LC46 at pH = 6.5 was $67 \pm 4$ nM, and $205 \pm 28$ nM at pH = 7.4, indicating stronger Myc-Max affinity at pH = 6.5[36]. The $K_D$ derived from single-molecule experiments was close to the SPR results ($K_D = 351 \pm 28$ nM, as shown in Supplementary Fig. 30). The average dwell times ($\tau_{bind}$ and $\tau_{diss}$) of bound and dissociated (encounter intermediate) states were derived through the single exponential fitting of the dwell-time distributions from three different devices (Supplementary Figs. 31 and 32, Supplementary Table 5), further generating corresponding kinetic parameters, such as rate constants ($k_{bind}$ and $k_{diss}$ in Supplementary Table 5). The activation energy of the binding and dissociation processes was fitted to the Arrhenius equation (Fig. 3e), from which $E_{a\text{-}diss} = 109 \pm 9$ kJ/mol and $E_{a\text{-}bind} = 71 \pm 11$ kJ/mol. The enthalpy and entropy changes of the dissociation process were obtained using the second law of thermodynamics ($\Delta H_{diss} = 24 \pm 3$ kJ/mol, $\Delta S_{diss} = 50 \pm 9$ J/(mol·K)) (Fig. 3f).

## Myc-inhibitor interaction

We also studied the binding process between LC46 and two previously reported c-Myc$_{370-409}$-binding inhibitors, 10074-A4[29–31,33], as well as PKUMDL-YC-1205 (hereafter, 1205)[29–32] (Fig. 4a, b). The experiments were carried out under pseudo-physiological conditions (inhibitor solution in 0.01× PBS, 5% DMSO, 37 °C, pH = 7.4). The Myc-modified device was first incubated in PBS with 50 µM of small molecule inhibitor (10074-A4 and 1205, respectively, Fig. 4c) at 37 °C for 30 min, followed by real-time electrical monitoring. Multi-state current signals were observed in the current trajectories (Fig. 4a, b). Control experiments indicated that the signals in Fig. 4 came from the interaction between LC46 and inhibitors (Supplementary Fig. 26b, c). In

comparison with free LC46, the inhibitor-bound complexes were more stable in a partially folded state. Interestingly, the current signals in solutions containing inhibitors were similar to those in the DQ47 solution, showing two obvious interaction patterns (clustered pulse signals and bistate-like signals). The fast-clustered pulse signals with higher conductance levels should result from the partial folding of LC46 in the presence of inhibitor. In comparison with the Myc* state formed with DQ47, the dominant bistate binding signals were observed at a much higher concentration of inhibitors due to their weaker binding strength[32], and the pulsed spike current signals remained. The encounter intermediate ensembles formed with 10074-A4 (Myc*$^1$) or 1205 (Myc*$^2$) could be derived from the above experiments (Fig. 4a, b).

To explore the interaction process between LC46 and small molecule inhibitors, concentration-dependent experiments were performed (Supplementary Figs. 33 and 34). With the increase of inhibitor concentration, the current signals gradually changed from fast-clustered pulse signals to bistate-like signals with pulsed spikes. As the presence of DQ47 led to a fully folded (more condensed) conformation of LC46, this similar change pattern in current signals indicated that these small molecules also had the potential to cause extended-to-condensed conformational changes of LC46. The population of encounter intermediate ensembles declined and the LC46-inhibitor bound state enlarged as the inhibitor concentration increased. However, because of the weaker affinity of the inhibitors, the pulsed spike signal remained at a high inhibitor concentration of 200 µM, implying the presence of transient encounter intermediates and inhibitor-binding conformational ensembles even at such a high inhibitor concentration. The current signal pattern observed in the inhibitor concentration-dependent experiments further verified the encounter-intermediate pathway proposed in the above Myc-Max experiments. Using the current distributions from real-time trajectories, the populations of the bound state were obtained at different inhibitor concentrations (Fig. 4d); thereby, dissociation constants were derived by fitting the Hill equation (Supplementary Table 6): $K_D$ (10074-A4) = $33 \pm 6$ µM, and $K_D$ (1205) = $16 \pm 2$ µM. These results were comparable to those from surface plasmon resonance experiments[32], showing that our strategy has great potential for use in IDP-targeted drug discovery.

## Competition between DQ47 and inhibitor

Competition for LC46 between DQ47 and small molecule inhibitors was monitored in real time (Fig. 5a–d). On the basis of the previous inhibitor dose-dependent experiments, 100 µM solution was used for both inhibitors. The real-time current trajectories in the competition experiments showed that the original stable bistate-like equilibria between LC46 and inhibitor were significantly disrupted by adding only 1 nM DQ47 solution, and fast-clustered pulse signals appeared (Supplementary Figs. 35 and 36). As the concentration of DQ47 increased, a lower current state began to appear, indicating that an unstable conformational ensemble intermediate appeared with a higher degree of helicity. With further increase of DQ47 concentration, the population of this new encounter intermediate ensemble increased until another bistate-like signal with a low conductance level appeared (Fig. 5b, d). Since neither 10074-A4 nor PKUMDL-YC-1205 have total charge, this bistate-like signal with a lower current level corresponded to the binding process between LC46 and DQ47 because of the more compact structure and the additional total positive charge carried with DQ47. At higher DQ47 concentrations, the lower bistate-like signal was more stable and even replaced those seen with the inhibitor (i.e., Myc*$^1$ or Myc*$^2$) (Supplementary Figs. 35g and 36g). At lower DQ47 concentrations, the two bistate equilibria were partitioned by clustered pulse signals during real-time monitoring (Fig. 5a–d), and there was obvious competition between different equilibria. The decisive factor in this competition lies in the encounter

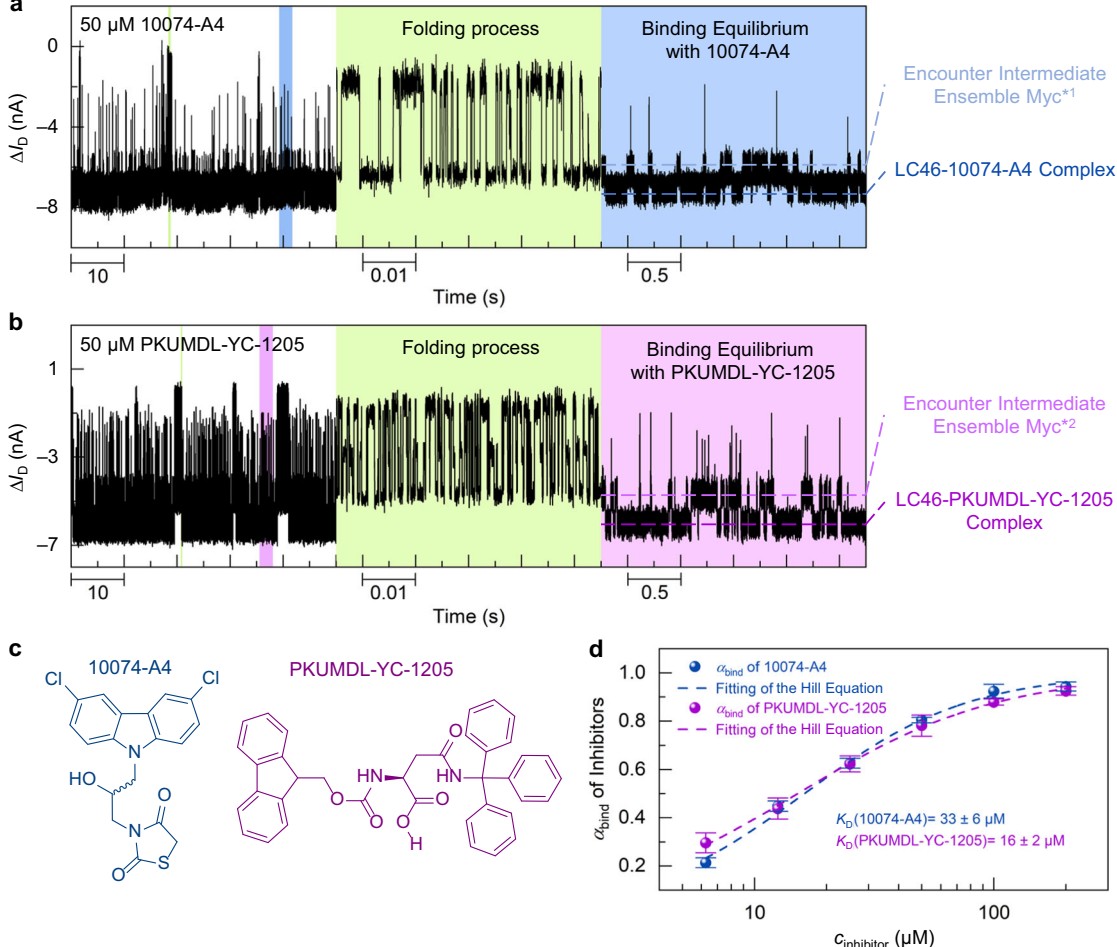

**Fig. 4 | Real-time monitoring of interactions between c-Myc and small molecule inhibitors. a, b** Real-time current trajectories (time resolution of 17 µs) in 50 µM 10074-A4 (**a**) and PKUMDL-YC-1205 (**b**) solutions (0.01× PBS, 5% DMSO, pH = 7.4). The left column is real-time current data over 60 s, the middle column is an enlarged image of the green-marked area, and the right column is an enlarged image of the blue- (**a**) or purple- (**b**) marked area. **c** Chemical structures of the

inhibitors 10074-A4 (blue, racemic mixture in the experiments) and PKUMDL-YC-1205 (purple). **d** Population of Myc-inhibitor binding states at different inhibitor concentrations ($n = 3$, number of the devices, data are presented as mean values ± SD). The dissociation constants of the inhibitors were obtained by fitting the Hill equation (Supplementary Table 6). Source data are provided as a Source Data file.

intermediate ensembles of LC46 that is formed according to the microenvironment of the solution (i.e., DQ47 or inhibitor) (Fig. 5e).

In comparison with the LC46-DQ47-only system, the bistate signal of the LC46-DQ47 interaction was disturbed by inhibitor (1205 or 10074-A4) in the solution, suggesting that a certain concentration of inhibitor could also interfere with the formation of Myc*, providing the possibility of specific encounter intermediate ensembles induced by inhibitors that were disfavourable for the heterodimerization of Myc with Max. Figure 5e shows the process of competition between DQ47 and the inhibitor (using 1205 as an example). With changes in the relative concentration ratio of DQ47 and 1205, LC46 tends to fold into different encounter intermediate ensembles, which further selectively bind to either DQ47 or the inhibitor.

To measure the effect of inhibitors on LC46-DQ47 binding affinity, the population of LC46-DQ47 binding states was calculated at different DQ47 concentrations in the presence of 100 µM inhibitor (Fig. 5f). The apparent dissociation constants of the LC46-DQ47 complex in the presence of the inhibitor were derived from the Hill equation (Supplementary Table 7), giving $K_D$ values of $3.5 \pm 2.6$ µM and $9.5 \pm 6.6$ µM with 10074-A4 and 1205, respectively. By comparing the apparent dissociation constants with the actual dissociation constant ($K_D$ [DQ47, pH 7.4] = $205 \pm 28$ nM), the presence of inhibitor hindered the specific binding between LC46 and DQ47. In addition, the competition

experiments showed the presence of different encounter intermediate ensembles upon binding to different ligands and again demonstrated the encounter-intermediate transition states for the Myc-ligand interactions.

## Discussion

We used high-performance SiNW-FET molecular nanocircuits to construct single-molecule c-Myc electrical biosensors. The device was capable of in situ monitoring of conformational transitions of the C-terminal IDR of Myc in real time, with high temporal resolution and sensitivity. In the absence of interacting molecules, the self-folding and unfolding of the Myc IDR were captured, implying the potential relationship between single-molecule dynamics and ensemble properties. The conformational ensembles with different dynamic behaviors were achieved to show the reconstruction of the energy landscape of the IDR. An encounter intermediate mechanism for the Myc-Max interaction was also revealed by real-time electrical monitoring, including a folding transition among conformational intermediates at low Max concentrations and a binding equilibrium at high Max concentrations. The $k_{obs}$ (the number of binding complex states per second) was plotted versus Max concentration to investigate the mechanism of Myc-Max (Supplementary Fig. 37). Although the $k_{obs}$ could be well fitted by a two-step induce-fit model[59], it remained unsure whether

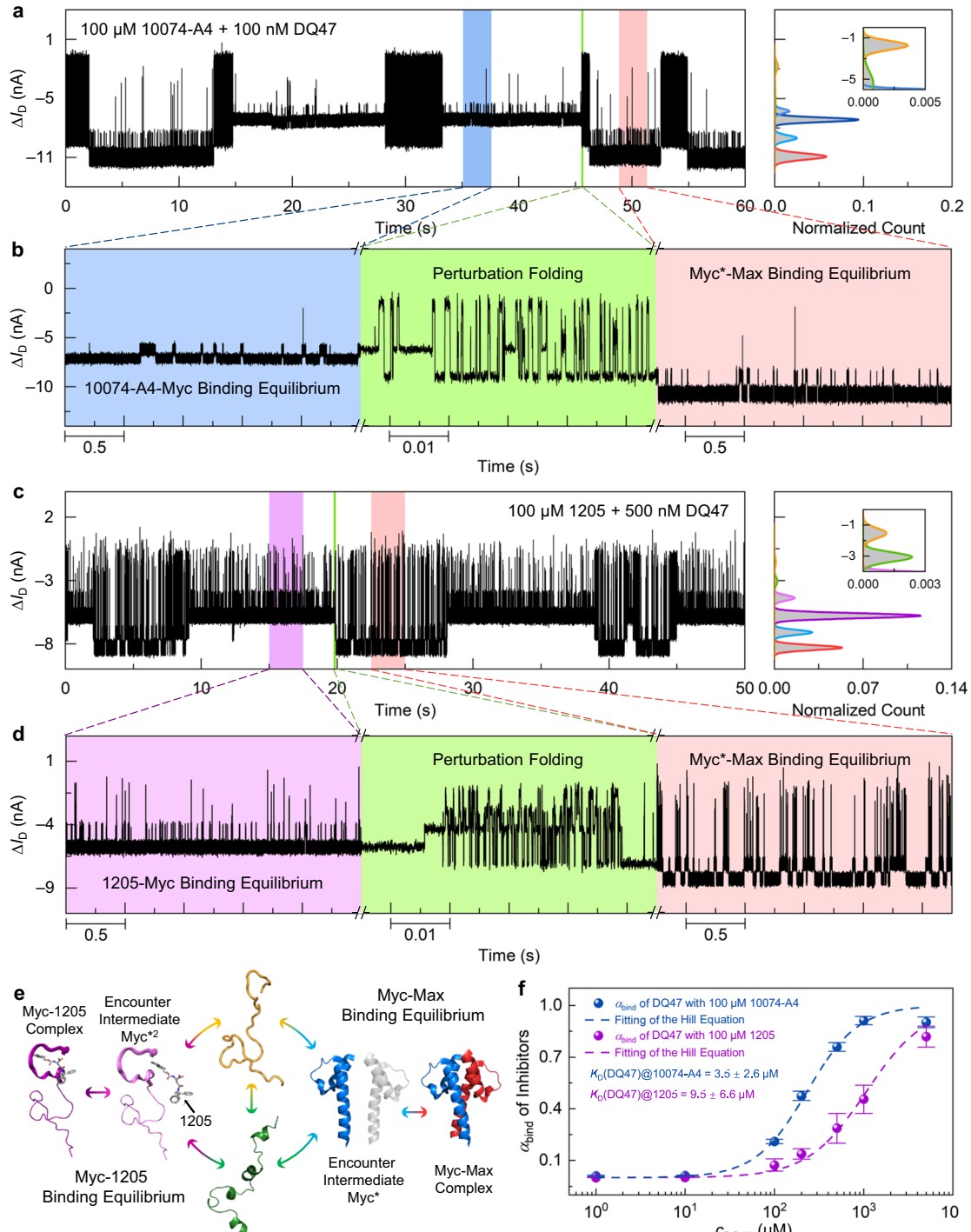

**Fig. 5 | Competitive experiments for Myc binding between small molecule inhibitors and Max. a, b** Real-time current trajectories (time resolution of 17 μs) of competition between Max and 10074-A4 (0.01× PBS, pH = 7.4, 100 μM 10074-A4, and 100 nM DQ47). **a** The left panel shows current data over 60 s and the right image is the current distribution histogram; **b** shows magnified images of the marked areas in (**a**). Blue, green and red represent 10074-Myc binding equilibrium, interacting folding and Myc-Max binding equilibrium. **c, d** Real-time current trajectories (time resolution of 17 μs) of competition between Max and PKUMDL-YC-1205 (0.01× PBS buffer, pH = 7.4, 100 μM 1205, and 100 nM DQ47). **c** The left panel shows current data over 60 s and the right image is the current distribution histogram; **d** shows magnified images of the marked areas in (**c**). purple, green and red

represent 1205-Myc binding equilibrium, interacting folding and Myc-Max binding equilibrium. **e** Schematic diagram of the competition process between DQ47 and PKUMDL-YC-1205. The structures were adapted from ref. 32. Purple and light purple represent Myc binding state with 1205 and Myc* with 1205. The rest colors are defined as in Fig. 1. **f** The population of Myc-Max binding states at different DQ47 concentrations in the presence of 100 μM inhibitor (blue and purple represent 100 μM 10074 and 1205 solution; $n = 3$, number of the devices, data are presented as mean values ± SD. The apparent dissociation constants of the Myc-Max binding process in different inhibitor solutions were obtained by fitting the Hill equation (Supplementary Table 7). Source data are provided as a Source Data file.

LC46 underwent an induced-fit mechanism or a conformational selection mechanism[59,60]. To the best of our knowledge, the encounter intermediate ensemble Myc* during the Max-binding process was captured experimentally for the first time. Furthermore, different encounter intermediate ensembles were observed in both Myc-inhibitor interaction and competitive reactions for Myc between inhibitor and Max, implying that changes in the encounter intermediate ensembles are the key to regulate the function of the Myc IDR.

We are firmly of the opinion that our SiNW-FET single-molecule platform can be applied immediately to a wide variety of label-free biodetections at single-molecule resolution, such as to bioreaction mechanisms, protein folding, targeted therapy, drug discovery, enzymatic activity, and single-molecule sequencing. In addition to these, the proven reliability and compatibility with current CMOS technologies promise the development of low-cost multiplex electrical devices or sensors for practical applications, such as accurate molecular and point-of-care clinical diagnostics.

## Methods

### Materials

Peptides and compounds: Samples of LC46 and DQ47 were synthesized by GL Biochem. Ltd., Shanghai, China, with a purity of more than 95%, which was confirmed by the supplier, using HPLC and MS (Supplementary Figs. 1–4). Compound 10074-A4 (Catalog number: STK834743) was purchased from Vitas-M Laboratory, Ltd, which is commercially available from TopScience Co. (Shanghai, China), with a purity of more than 90%. The company provided $^1$H NMR. The 1× PBS buffer was ordered from M&C Gene Technology (Catalog number: CC008), which contains 8 g/L NaCl, 0.2 g/L KCl, 1.44 g/L $Na_2HPO_4$ and 0.24 g/L $KH_2PO_4$ at pH 7.4. Compound PKUMDL-YC-1205 used in the experiments was purchased from Shenzhen Biochemilogic Technology Co. Ltd. The company provided NMR and LC-MS.

Characterization: The purity of PKUMDL-YC-1205 and 10074-A4 was rechecked by performing HPLC, MS, $^1$H and $^{13}$C NMR (Supplementary Figs. 5–12). Both compounds are over 90% pure, which was determined by an Agilent 1206 Infinity high-performance liquid chromatography instrument with an SB-C18 column (4.6 mm × 150 mm, 5 μm) with methanol and water with 0.1% formic acid as the mobile phase. The flow rate was 1 mL/min, and the peak was detected at 254 nm. $^1$H NMR spectra were recorded on a Bruker 500 MHz spectrometer. $^{13}$C NMR spectra were recorded on a Bruker 400 MHz spectrometer. The chemical shift values ($\delta$) are reported in ppm relative to tetramethylsilane as the internal standard. High-resolution mass spectra were recorded on a Bruker Solarix XR FTMS mass spectrometer using ESI (electrospray ionization).

### Device fabrication and characterization

SiNW growth: Gold nanoparticles (AuNPs, Sigma-Aldrich, an average diameter of ~10 nm) were dispersed on silicon wafers (300 nm thermal oxide layer) as catalysts. P-type SiNWs were synthesized through chemical vapor deposition in a tubular furnace at 460 °C for 25 min. Here, 2.5 sccm $Si_2H_6$ (Matheson Gas Products, 99.998% Purity) was used as the reactant gas, 0.30 sccm $B_2H_6$ (100 ppm, diluted in $H_2$) as a p-type dopant (B/Si ratio about 1/100000), and 7.0 sccm $H_2$ as the carrier gas (Supplementary Fig. 13).

Device fabrication: SiNWs were transferred to a silicon substrate with a thermal oxide layer using mechano-sliding[50,52]. Electrode patterns were defined using UV lithography. The oxide shell of SiNWs was removed by wet-etching with an HF-$NH_4F$ solution (40% $NH_4F$: 40% HF, 7:1). Metal electrodes (8 nm Cr and 80 nm Au) were deposited by thermal evaporation. A protective $SiO_2$ layer (30 nm) was deposited via electron beam evaporation. After lift-off, the SiNW-FET device array was obtained (Supplementary Figs. 13 and 14). The SiNW-FET area remained exposed for subsequent experiments, while the rest was covered with a SU-8 protective layer.

Electrical characterization of SiNW-FET devices: The boron-doped Si substrate was employed as the global bottom gate with a 1000 nm $SiO_2$ layer as the dielectric layer. SiNW-FET devices were located on a Karl Süss (PM5) manual probe station, and the electrical characterization was then conducted at room temperature using an Agilent 4155C semiconductor analyzer. SiNW-FETs showed the typical p-type behaviors with good Ohmic contacts (Supplementary Fig. 15). A platinum probe was also used as a liquid gate electrode with a droplet of PBS buffer solution on the device as the dielectric layer.

### The modification of single Myc peptide

Confinement reaction window: A gap-opening procedure was employed to create the confinement reaction window. The devices were first spin-coated with a PMMA layer (950, A4) on the surface, followed by baking at 180 °C for 2 min. Electron beam lithography (EBL) was used to create a design line pattern of approximately 5 nm width at a specific position, generating a window precursor (Supplementary Fig. 13). The resist was developed with a water/isopropanol mixture (V:V = 1:3) at 4 °C for 1 min, aided by sonication. After development, the devices were rinsed with deionized water and dried with $N_2$ gas. A nanogap was created in the reactive region of the SiNW by wet-etching in an HF-$NH_4F$ solution (40% $NH_4F$: 40% HF, V:V = 7:1) for 5 s. Subsequently, the wafers were washed, dried, and sealed in a balloon flask with 10-undecynoic acid powder. The hydrosilation reaction was conducted at 90 °C for 12 h under an Ar atmosphere. After cooling, the device wafers were washed sequentially with dichloromethane, acetone, and deionized water.

Single LC46 modification: The wafers were immersed in an MES buffer solution (0.05 M, pH = 6.5) containing NHS (20 mM) and EDC (10 mM) for 1 h at room temperature. After thorough washing with deionized water and drying with $N_2$ gas, the wafers were then immersed in a 2-Maleimidoethylamine hydrochloride solution (10 mM in DMF) for 2 h. Following thorough washing with DMF and drying with $N_2$ gas, the wafers were exposed to a solution of LC46 (5 μM in PBS buffer, pH = 7.4, 5% DMSO) for 12 h at 4 °C. Subsequently, the devices were rinsed with PBS buffer (pH = 7.4, 5% DMSO) and dried using $N_2$ gas.

### Characterization of single-Myc modification

Fluorescent modification: To characterize the functionalized device with single-Myc modification, FITC isomer I was used as the fluorescence group. The Myc-modified device was immersed in a 10 nM FITC solution in PBS (pH = 8.0, 5% DMSO) and reacted for 12 h at 4 °C. The FITC reacted with the lysine amino group on the IDR sequence, resulting in fluorescence labeling covalently. After rinsing with pH 8.0 PBS buffer and drying with $N_2$ gas, the FITC-labeled device was prepared for fluorescence characterization.

Optical characterization: Covered with a PBS buffer liquid layer and a coverslip, The device was placed on a microscope objective stage. STORM was performed using a Nikon Ni-E microscope with a ×100 objective lens. The FITC-modified molecules on the device were excited by a 480 nm laser, and the emitted signals were recorded using an EMCCD with a 50-ms exposure resolution. Fluorescent signals at the target location were recorded for 5 min. The optical images were reconstructed and analyzed using Advanced Research software. Single fluorescence spots were observed on individual SiNW-FET devices, confirming successful modification (Supplementary Fig. 17).

AFM characterization: The non-FITC-modified device was characterized with AFM (AFM, Bruker AFM Dimension Icon). The AFM image (Supplementary Fig. 18) was generated at the ScanAsyst mode with a sampling rate of 1.00 Hz and 512 samples per line, which confirmed the presence of a single LC46 attached to the etched gap on the side of silicon nanowires.

SPR experiments: SPR experiments were performed at 25 °C using a Biacore T200 (GE Healthcare Biacore, Uppsala, Sweden)

instrument. Before the direct binding assays of DQ47, N-terminal biotinylated-Myc peptide (synthesized by GL Biochem Ltd.) was immobilized onto a SA chip (GE Healthcare) to a level of approximately 150 response units (RU). The experiments were performed in 1 × PBS-P (GE Healthcare) with 5% DMSO. The certain concentration gradient DQ47 samples were applied over the surface at 30 μL/min for 120 s association time with a dissociation time of 240 s. The raw data obtained was analyzed with the Biacore T200 Evaluation Software 2.0 (GE Healthcare). The $K_D$ values were calculated from the kinetic binding model fitting of the response-concentration plot to the equilibrium curves. The results are shown in Supplementary Fig. 30 and Supplementary Table 2.

### Real-time electrical measurements

The wafer with single protein-decorated devices was enclosed in a PDMS cube with a small reaction chamber. A 50 μL protein solution of a certain concentration was added to the microchamber. Precise temperature control was achieved using an INSTEC hot/cold chuck with a proportion-integration-differentiation control system and liquid nitrogen cooling. Real-time electrical measurements were conducted with a source-drain bias of DC 300 mV and a gate bias of 0 mV, using an HF2LI Lock-in Amplifier. The source-drain current was amplified by a DL1211 preamplifier and collected by the HF2LI Lock-in Amplifier with a 10 kHz low-pass filter. Three different devices to ensure signal reproducibility (Supplementary Fig. 31).

### Statistics and reproducibility

Statistical analysis was conducted relying on the original data without randomization and blinding treatments. The experiments have separately proceeded in three independent devices to manifest the reproducibility.

### Reporting summary

Further information on research design is available in the Nature Portfolio Reporting Summary linked to this article.

## Data availability

All data that support the findings of this study are available within the main manuscript and the supplementary files. All the datasets used in this work are available online from the Zenodo repository at https://zenodo.org/record/8188411 (https://doi.org/10.5281/zenodo.8188411). The protein structure (1NKP) used in this work is from the RCSB PDB with the ID "1NKP [https://doi.org/10.2210/pdb1NKP/pdb]". Source data are provided with this paper.

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

## Acknowledgements

We acknowledge primary financial support from the National Key R&D Program of China (2021YFA1200101 and 2022YFE0128700) to X.G., the National Natural Science Foundation of China (22150013 and 21933001 to X.G., and 22237002 to L.L.), the New Cornerstone Science Foundation through the XPLORER PRIZE to X.G., "Frontiers Science Center for New Organic Matter" at Nankai University (63181206) to X.G., and the Natural Science Foundation of Beijing (2222009) to X.G.

## Author contributions

X.G. and L.L. conceived and designed the experiments. W.L., D.Y., and J.F. fabricated the devices and performed the device measurements, with contributions from Z.Y. L.C. and Q.S. prepared the protein samples and performed molecular dynamic simulations. W.L., D.Y., L.C., X.G. and L.L. analyzed the data and wrote the paper. All the authors discussed the results and commented on the manuscript.

## Competing interests

The authors declare no competing interests.
