## [Peer Review File · Nature Communications]

REVIEWER COMMENTS

Reviewer #1 (Remarks to the Author):

This is a very interesting manuscript that describes the use of a silicon nano wire functionalized with a single molecule of an intrinsically disordered protein (c-Myc). Through measurements of current, the authors study the conformational ensemble of c-Myc as well as its interaction with a binding partner (Max) and available inhibitors. The authors clearly demonstrate the sensitivity of this approach to probe different conformational ensembles of the free IDP, as well as its structural transitions in the presence of binding partner and inhibitors. Overall, I think the work provides an important contribution to the IDP field. I am in favor of publication in Nature Communications, if the following points can be addressed:

1) Page 6: The authors provide evidence for a single-LC46 functionalization by attaching FITC for fluorescence characterization. It was not clear how this method can unambiguously determine that the devices were modified with a single LC46 molecule, when the LC46 sequence contains six lysines. This aspect needs to be clarified, also in terms of the achievable resolution for the fluorescence characterization.

2) Page 7: The authors assign the observed decrease in current to partially or transiently folded states of LC46 induced by positive charges coming closer to the SiNW. Would it be possible to provide some more evidence for this using additional experiments modulating the screening of charges (using different salt concentrations for example)? The charge distribution of LC46 is almost uniform, meaning that the positively charged residues (K/R) are interspersed with negatively charged amino acids (D/E). Globally, LC46 has a charge of +3 (or only +2, if the NH at the N-terminus of the protein is not counted). On this basis, it is difficult to understand how this charge is the only parameter modulating the current. It would be important to understand in more detail the physical origin of the observed current changes, in order to provide evidence for a general applicability of the method to other IDP systems (with varying pIs and sequence compositions).

3) Related to point 2: The attachment to the SiNW is done via a cysteine residue at the C-terminus of the protein construct. Looking at the structure of the c-Myc-max complex, this residue would be right at the end of the folded structure. I understand the need to have proximity of the protein to the nanowire to facilitate measurements (changes in current), but this close attachment could significantly perturb the physiological relevance of the interaction measurements with max. Could the authors comment on this?

4) Page 8: The authors state “The short duration of these clustered signals in our experiments was in accordance with the predominantly disordered nature of LC46”. Some references to literature should be

added to support the claim that LC46 is mostly disordered for example from NMR experiments or maybe small angle X-ray scattering. Are such data available?

5) Page 8-9: Would it be possible to relate the observations in Fig. 2c/d to more structural characteristics of the c-Myc ensemble? The current interpretation is that two conformational ensemble intermediates are observed, but what do these ensembles correspond to (transiently formed alpha-helices, hydrophobic collapse, molten globule etc)? How can the observed temperature dependence be explained (taking into account thermodynamic considerations)?

6) Page 11: The authors determine the dissociation constant for the Myc-Max complex (Fig. 3d). Are these KD values in agreement with literature reports?

7) It was not clear to which extent kinetic information can be extracted (koff and kon rates for the c-myc/max complex).

Minor points:

- The legend to figure 1b should be completed to add explanations of the different subpanels.

- Please define the abbreviation "SiNW-FET" at the first place of use (not only in the abstract)

- Figure 1c/d: It is not clear enough from the figure what is meant by "induction". The color code is well-respected for disorder (yellow) and self-folding (green). Should the gray structure in panel d, actually be magenta to follow the logic in panel c?

- What is the reason for using a DMSO-containing buffer for the experiments? Is that for comparability with the measurements using inhibitors? I am asking because some proteins may not be compatible with these conditions.

- Please add temperature to the experiments in Fig. 2a.

Reviewer #2 (Remarks to the Author):

The manuscript describes the construction of a single-molecule electrical nanocircuit functionalised by the intrinsically disordered region of the cancer-associated protein c-Myc and its use to study conformational transitions in c-Myc either in its apo state or in response to binding to its binding partner Max as well as binding to a small molecule inhibitor. Overall, the manuscript is well written and the data figures are appropriate.

Major points:

- 1) A major conclusion of the manuscript relates to the potential identification of a pre-binding state. However, the interpretation of the "additional" experimental signal as a pre-binding state is questionable. In addition, even if this would be a pre-binding state the manuscript/data provide little insight into the nature of this pre-binding state
- 2) The authors only focus on induced fit as a model for explaining their experimental data and completely neglect conformational selection, which is a bit surprising given previous evidence of the presence of a transient α -helical region in the IDR of c-Myc prior to binding.

Reviewer #3 (Remarks to the Author):

This manuscript by Guo, Lai, and co-workers describes exceptionally creative and ground-breaking single molecule studies. Intrinsically disordered proteins (IDPs) would appear to be poor subjects for single molecule experiments, as random conformational changes would frustrate analysis. However, perhaps through clever choice of a target subject, the authors may have demonstrated the repeatable motions of an IDP in a number of interesting states, including control, binding to ligand, and binding to ligand in competition with an inhibitor. For the following reasons, however, this reviewer is not convinced that the authors are observing what they claim.

1. The agreement in the HRMS characterization between theoretical and observed MWs is suspiciously high. Such agreement might happen 5% of the time, but every molecule with a tiny difference in the submilli-amu seems too good to be true. I do not believe the authors have made the molecules they claim to have made and suggest that they use the checklist for characterization required for organic chemicals by organic chemistry journals (e.g., JOC). In other words, I'd like to see a fully assigned ^1H and ^{13}C NMR spectra and LC-MS to check purity, etc. Without such data, the reader cannot believe the results as presented.

2. The authors claim that only one protein is attached per SiNW-FET. However, they show this using a FITC labeling of the protein and then imaging to observe only one label. First, FITC labelling is not quantitative. The conditions could label a low percentage of proteins that are present. Second, FITC is readily photo-bleached, which again would lower the yield of active FITC labels present. The authors could use in-liquid AFM to rigorously demonstrate single molecule attachment. Or they could moderate the manuscript's claims.

3. To be blunt, the data looks too clean for single molecule data. I've looked at a lot (a lot!) of single molecule data in my lab and others. The histograms do not look so tight and there's considerably more spread in the signals observed. For an IDP, I'd expect even more spread, not less, as one could imagine a number of variant conformations accessed by a less tightly defined structure. Has this data been subjected to anti-aliasing or other signal processing to force it into two or three levels?

4. Would the authors also please discuss the time resolution of their experiments? I believe the "instantaneous" rise times are not due to instant transitions, but due to the lower time resolution of their experiments than the time required to move between conformational states. However, it would be important to clarify this point for the readers.

Minor issues:

1. What are the dimensions of the SiNW-FET? What is the composition of the PBS used (PBS can vary somewhat amongst different investigators)?

p. 3 of SI: catalog is misspelled.

Reviewer #4 (Remarks to the Author):

In this manuscript, the authors reported dynamic interactions between an intrinsically disordered protein, c-Myc and its binding partners (Max/inhibitors) using a single-molecule nanocircuits approach, where the single protein is attached to the silicon nanowire field-effect transistor (SiNW-FET) via linker molecules. The authors observed current fluctuations between multiple current levels during dose-dependent measurements and claimed that those current levels correspond to Myc's intermediate conformational states including disordered state, partially folded state, fully folded state, and bound state with either Max or inhibitors. In addition, the authors showed temperature dependence of a few conformational states, which could be responsible for the self-folding process. However, the conclusions

drawn from the results and discussion reported in this manuscript are based solely on the simple assignment of current fluctuations to potential conformational states of the protein without any direct and/or supporting validation (e.g., crystal structure, bulk assay, alternative single molecule measurements such as FRET or EPR (DEER)). In addition, this reviewer finds that the manuscript lacks new information or insight into the mechanisms of protein folding and protein-protein binding. Further details are outlined below.

1. What would be the difference between the binding-induced-folding described here and the induced-fit model?

2. The device schematic image (Fig 1a) is not scaled, which may confuse readers. The authors need to provide more information on the dimension of the nanogap and the scale of LC46? If the nanogap is too large, DQ47/inhibitors could interact directly with SiNW, regardless of the presence or absence of LC46, leading to similar random telegraph signals. If the nanogap is tight, the LC46 would be confined in the nanogap, limiting its conformational motions and increasing its potential to be sticking to the SiNW or Su-8 if it is in a disordered, long chain-like state as depicted in Fig1a.

3. The authors need to explain the method used to control the attachment of a single protein. What was the yield? Also, was the attachment stable for the long-duration measurement? There could be dissociation, significant degradation, unfolding or misfolding during the long-term, temperature-dependent measurements. It is important to know how many devices and proteins were tested in the manuscript, and confirm that the signals are reproducible and the dwell times are with different binding partners and temperatures.

4. There are no details on stochastic optical reconstruction (microscopy) measurements performed on the Silicon devices, which are crucial to demonstrate a single protein attachment.

5. The electronic characteristics of SiNW FETs, such as the IV and IVg, should be presented here so that the readers can assess the gating effects of small protein charges.

6. (Page7, line 10). The authors need to specify the positive charges on the surface of LC46 and their location relative to the SiNW, and if they are within the Debye length. What is the Debye length in this work? How many charges contribute to signal generation? Also, how much protein charges can induce or modulate entire NW conductivity? The author could test different pH, buffers, and ionic strengths to validate signal generation. Also, the authors could repeat these experiments with selective mutation of LD46 (different charges) to verify the signal direction and amplitude.

7. The electronic signals presented in all figures appear to be simple 1-dimensional motions of the protein, however, if LD46 can move and rotate freely, its conformations would not remain consistent. Thus, the gating effects of the protein would change during the conformational transition.

8. The authors need to provide more data or compelling arguments for the transient coiled and partially folded state (Fig.1). How do the authors determine and know each current state as partially folded, transient induction state, folded state, or bound conformational state? These signals could potentially be the results of a transition between multiple partially folded states or binding/unbinding to the device surface or SiNW. Additionally, what is the implication and role of the partially folded and pre-folding conformation? The authors need to elaborate on these.

9. (Page 9, lines 1-3). It is not obvious that 1* is a dynamic self-folding process? Why does the protein undergo a self-folding process at low and high temperatures? The authors need to provide a more convincing argument and experimental support for this.

10. The authors need to clarify what “n” represents in the Figures. Is it the number of devices, proteins, or signal segments?

11. Similar to the previous point, the authors need to provide details on the charges and charge distribution of DQ47, and they get close to the SiNW when bound? How does binding result in substantial changes in the conductance of the devices?

12. How do the authors know that DQ47 induces induction of LC46 without direct interaction? Need more data or convincing arguments.

13. This manuscript shows multiple conformational populations of LC46, and thus it could be feasible to determine these conformational states via crystal structure analysis. Are there any crystal structures of LC46 or LC46-DQ47 complexes available to support the findings in this manuscript?

14. The temperature dependence of intermediate states, binding, and dissociation constants could be further validated by FACS or fluoresces assaying experiments, which will provide stronger evidence.

15. The LC46-inhibitor complexes show similar changes in current direction and amplitude (dI) as the LC46-DQ47 complex, indicating that both inhibitors have the same level of positive charges of DQ47. Why/how does the current drop for the LC46-inhibitor complex? What are the charges for 10074 and

1205 and how the charges are determined? To verify the signal direction and amplitude, the authors could repeat these experiments using synthetic compounds with additional positive or negative charges.

16. (Figure 4, 5). In the presence of Max and inhibitors, the magnitude and levels of current signals were changed. i.e., current levels of LC46-Dq47 in Fig3a vs. Fig5, LC46-10074 in Fig4a vs Fig5, and LC46-1205 in Fig4b vs Fig5. How do the Max and inhibitor affect the conformational states of LD46? Why do current levels change for the same complex? (i.e., the magnitude of the bound current level for the LD46-DQ47/10074/1205 should be the same?).

17. Since this approach allows long-term monitoring of protein fluctuations, it would be nice to see the dynamic disorder of protein fluctuations (e.g., dwell times vs. time, color-coded enzyme behaviors vs time, [dx.doi.org/10.1021/ja311604j](https://doi.org/10.1021/ja311604j), J. Am. Chem. Soc. 2013, 135, 7861–7868, [dx.doi.org/10.1021/ja211540z](https://doi.org/10.1021/ja211540z), J. Am. Chem. Soc. 2012, 134, 2032–2035).

Point-to-point response to the reviewers' comments (in blue)

We thank the four reviewers for their constructive comments and suggestions. We have revised our manuscript accordingly. Here is a brief summary of the major changes in the revised manuscript followed by the point-to-point response.

1. We have added more detailed information about the STORM method as suggested by Reviewers #1, #3 and #4. The AFM results were provided to further verify the single-LC46 modification of the device. The information of the AFM experiments was added in the Supplementary Information. Please see Page 6 in the main text, the “Device fabrication and characterization” section and Supplementary Fig. 18 in the Supplementary Information.
2. We have added the SPR results of the biotinylated-Myc and Max peptides' binding for a complemented reference of the dissociation constant K_D following the comments of Reviewer #1. Relevant information of the experiments and descriptions have been added to Supplementary Information and the main text. Please see Page 12 in the main text and the “Device fabrication and characterization” section and Supplementary Fig. 30 in the Supplementary Information.
3. We have revised Fig. 1 for a better understanding according to the question of Reviewer #1. Please see Page 19 in the main text for revised Fig. 1.
4. After further consideration on the mechanism, the term, “pre-binding intermediate”, has been revised to “encounter intermediate ensemble” for a more accurate expression according to the comments from Reviewer #2. Please see Pages 2, 4, 5, 10, 11, 12, 13, 14, 15, 16, 17 and Figs. 1 and 3 in the main text.
5. After further consideration on the mechanism, references about conformation selection model have been added into our manuscript. The term, “binding-induced-folding”, has been revised for a better understanding according to the comments from Reviewers #2 and #4. Relevant descriptions and discussions for the mechanism have been added in the main text. A kinetic analysis was also made for the discussion of binding mechanism. Please see Pages 4, 10, 11, 13, 14, 16 and 17 in the main text and Supplementary Fig. 37 in the Supplementary Information.
6. The fully assigned spectra of HPLC, $^1\text{H-NMR}$, $^{13}\text{C-NMR}$ and MS results of the two inhibitor molecules as well as HPLC and MS results of two peptides used in our experiments have been provided in the Supplementary Information for a better understanding according to the comments from Reviewer #3. The information of

the spectra has also been revised. Please see “S1. Materials” section and Supplementary Figs. 1–12 in the Supplementary Information.

7. The time resolution of our experiments has been explained in the legends of the figures. Necessary annotation and revision have been made for each figure in the main text for a better understanding according to the comments from Reviewer #3. Please see Page 8 and Figs. 1–5 in the main text.
8. The dimension information of LC46 modified device and the results of control experiments using molecule linkage devices without LC46 modification have been provided for a better understanding on the current signals according to the comments from Reviewer #4. Relevant descriptions have added in the main text and Supplementary Information. Please see Pages 7, 9, 10 and 13 in the main text, and Supplementary Figs. 21 and 26 in the Supplementary Information.
9. Experiments measuring different devices have been added to show the reproducibility of the Myc*-Max binding signals. Information about the yield and the stability of devices, as well as reproducible signals have been added in the main text and the Supplementary Information. Please see Page 12 in the main text, as well as the “Device fabrication and characterization”, “Real-time current measurements and dynamic analysis” section and Supplementary Fig. 31 in the Supplementary Information.
10. The electronic characteristics of SiNW FETs have been added in the revised manuscript for a better understanding of the working principle of the devices following the comments from Reviewer #4. Please see Page 6 in the main text, and the “Device fabrication and characterization” section and Supplementary Fig.15 in the Supplementary Information.
11. The results of PBS concentration-dependent experiments have been added in the revised manuscript to see the influence of ionic strength on the device following the comments from Reviewer #4. The information of the PBS has also been provided for a better understanding. Please see Pages 7, 9, 13 and Figs. 2, 4, 5 in the main text, and Supplementary Fig. 20 and Supplementary Table 1 in the Supplementary Information. Descriptions of the Supplementary Figures have also been added to the legend of each figure containing real-time trajectories.
12. The descriptions of state 1* have been revised for a better understanding of the self-folding process following the comments from Reviewer #4. Please see Fig. 2 and Page 9 in the main text.

13. “n” in the manuscript has been clarified according to the comments from Reviewer #4. Please see Figs. 1-5 in the main text.
14. The descriptions about the LC46-inhibitor interaction have been added for a better understanding according to the comments from Reviewer #4. Please see Pages 13 and 15 in the main text.
15. The dynamic disorder analysis has been added in the revised manuscript to measure the conformation fluctuation of the encounter intermediate ensemble according to the comments from Reviewer #4. Please see Page 11 in the main text and Supplementary Fig. 28 in the Supplementary Information.

Point-to-point response to the reviewers' comments (in blue)

Reviewer #1 (Remarks to the Author):

Comments:

This is a very interesting manuscript that describes the use of a silicon nano wire functionalized with a single molecule of an intrinsically disordered protein (c-Myc). Through measurements of current, the authors study the conformational ensemble of c-Myc as well as its interaction with a binding partner (Max) and available inhibitors. The authors clearly demonstrate the sensitivity of this approach to probe different conformational ensembles of the free IDP, as well as its structural transitions in the presence of binding partner and inhibitors. Overall, I think the work provides an important contribution to the IDP field. I am in favor of publication in Nature Communications, if the following points can be addressed:

1) Page 6: The authors provide evidence for a single-LC46 functionalization by attaching FITC for fluorescence characterization. It was not clear how this method can unambiguously determine that the devices were modified with a single LC46 molecule, when the LC46 sequence contains six lysines. This aspect needs to be clarified, also in terms of the achievable resolution for the fluorescence characterization.

Response: We thank the reviewer for the comment. In order to verify the single-LC46 functionalization with fluorescence methods, we introduced FITC as the fluorescent group. Since there are multiple lysine reactive sites in the LC46 sequence, we carried out the reaction at a low FITC concentration to reduce the reactivity. We utilize stochastic optical reconstruction microscopy (STORM) to realize fluorescent characterization at the single-molecule scale. STORM records the stochastic switching signals of fluorescent molecules within a field of view and localizes individual molecules by reconstruction from numerous images accumulated over time, allowing a resolution of 20–30 nm by avoiding overlapping of fluorescence molecules within the same diffraction limited volume (*Science* **2018**, *361*, 880). The single-molecule images in Fig. 1b and Supplementary Fig. 17 showed single localization points with a resolution of ~20 nm reconstructed from switching fluorescence signals from FITC, demonstrating single-LC46 modification. In order to further validate this conclusion, we conducted AFM experiments (**Fig. R1**) to verify the single-LC46 modification.

Fig. R1 | AFM image of a single-LC46 modified SiNW-FET device. a) AFM image of a SiNW device, where a single LC46 was attached on the surface of a SiNW. b) The height distribution of different sections in the AFM image (blue for bare SiNW and red for the locus of protein attachment). The difference of the height is about 7 nm, which is consistent with the size of LC46 (~5.3 nm) and the molecule linkage (~1.7 nm).

Our Revision: We have added **Fig. R1** as **Supplementary Fig. 18** in the revised Supplementary Information. Relevant descriptions about STORM and AFM image have been added on **Page 6 in the revised main text**:

Fluorescence images and stochastic reconstruction results showed a single location spot with a resolution of ~20 nm, corresponding to a single fluorescence molecule (Fig. 1b and Supplementary Fig. 17). The device without FITC modification was also characterized using atomic force microscopy (Supplementary Fig. 18), verifying that the device was successfully modified with a single LC46 molecule.

More detailed descriptions about the AFM and optical characterization have been added on **page 14 in the revised Supplementary Information** as below:

A PBS buffer liquid layer was added to the surface of the LC46-modified SiNW FET device firstly. The device was then covered by a coverslip and placed on the microscope objective stage. A Nikon Ni-E microscope with a $\times 100$ objective lens was positioned in close contact with the coverslip on the device through the lens oil. The field of view of the stochastic optical reconstruction microscopy (STORM, N-STORM Nikon) was zoomed on the modified device. Then, the FITC-modified molecules on the devices were excited by a laser of 480 nm, and an EMCCD (Andor) was used to record the emitted signals with a 50-ms exposure resolution. The switching fluorescent signals of the target location were recorded within 5 min. Subsequently, the reconstruction and analysis of the optical images were carried out with the Advanced Research software⁴. Only a single fluorescence spot was reconstructed on individual SiNW-FET devices (Fig. 1b

and Supplementary Fig. 17). The device without FITC modification was also characterized using atomic force microscopy (AFM, Bruker AFM Dimension Icon). The AFM image (Supplementary Fig. 18) was generated at the ScanAsyst mode with a sampling rate of 1.00 Hz and 512 samples per line, which confirmed the presence of a single LC46 attached to the etched gap on the side of silicon nanowires.

2) Page 7: The authors assign the observed decrease in current to partially or transiently folded states of LC46 induced by positive charges coming closer to the SiNW. Would it be possible to provide some more evidence for this using additional experiments modulating the screening of charges (using different salt concentrations for example)? The charge distribution of LC46 is almost uniform, meaning that the positively charged residues (K/R) are interspersed with negatively charged amino acids (D/E). Globally, LC46 has a charge of +3 (or only +2, if the NH at the N-terminus of the protein is not counted). On this basis, it is difficult to understand how this charge is the only parameter modulating the current. It would be important to understand in more detail the physical origin of the observed current changes, in order to provide evidence for a general applicability of the method to other IDP systems (with varying pIs and sequence compositions).

Response: We thank the reviewer for the constructive comment. According to the Debye screening effect, the electrical double layer (EDL) formed in ionic solution around the surface of silicon nanowires screens the charge signals from the solution environment beyond the Debye length, including random collisions of charged ions and charged molecules (*Nano Lett.* **2012**, *12*, 5245; *Nano Lett.* **2015**, *15*, 2143). Located within the EDL, the distribution of the charged molecules can generate the effective gating voltage to change the conductivity of SiNWs. In this study, the conductance changes of SiNWs originate from the difference of the charged site distribution caused by the conformational changes of the peptide within EDL (rather than the influence of a specific charged residue). During the conformational change of disordered peptides, the spatial distribution of the charged residues changes, leading to different charge distributions around SiNWs and different effective gating voltages (in this case, the total charge is positive). The coiled conformation of the peptide brought about a more compact positive charge distribution and a more positive gating voltage, resulting in the decrease of the conductance of SiNWs. Therefore, we believe that this strategy is universal and can be applied to different biomolecular systems, which produce the charge distribution changes during conformational changes (Please see examples: *Adv. Sci.* **2021**, 2101383; *Nat. Commun.* **2023**, *14*, 552).

3) Related to point 2: The attachment to the SiNW is done via a cysteine residue at the C-terminus of the protein construct. Looking at the structure of the c-Myc-max complex, this residue would be right at the end of the folded structure. I understand the need to have proximity of the protein to the nanowire to facilitate measurements (changes in current), but this close attachment could significantly perturb the physiological relevance of the interaction measurements with max. Could the authors comment on this?

Response: Thank you for this nice comment. In this work, the LC46 molecule was not directly linked onto the SiNW surface, but connected with a linkage molecule, that is 10-undecynoic acid. The linkage molecule was relatively flexible (rotation of C-C single bonds) and provided an about 2-nm space between the SiNW and the end of LC46, therefore reducing the perturbation caused by the fixation of the terminal of LC46 to some extent.

4) Page 8: The authors state “The short duration of these clustered signals in our experiments was in accordance with the predominantly disordered nature of LC46”. Some references to literature should be added to support the claim that LC46 is mostly disordered for example from NMR experiments or maybe small angle X-ray scattering. Are such data available?

Response: We thank the reviewer for the suggestion on. According to the NMR results using the δ 2D method, the free c-Myc₃₅₂₋₄₃₇ showed intrinsic disordered nature in contrast to the results of the ordered structure of the Myc-Max complex (*Biochemistry* **2019**, 58, 3144; *PLOS. Comput. Biol.* **2013**, 9, e1003249). In addition, the CD spectra (*Sci. Rep.* **2016**, 6, 22298.) also showed that the apo state of c-Myc₃₇₀₋₄₀₉ (6 amino acids shorter than LC46) was disordered.

Our Revision: We have added references to the relevant descriptions **on Page 8 in the revised main text:**

The short duration of these clustered signals in our experiments was in accordance with the predominantly disordered nature of LC46^{20,32,33}.

[20] Sammak, S. *et al.* Crystal structures and nuclear magnetic resonance studies of the apo form of the c-MYC:MAX bHLHZip complex reveal a helical basic region in the absence of DNA. *Biochemistry* **58**, 3144-3154 (2019).

[32] Jin, F., Yu, C., Lai, L. & Liu, Z. Ligand clouds around protein clouds: a scenario of ligand binding with intrinsically disordered proteins. *PLOS Comput. Biol.* **9**, e1003249 (2013).

[33] Yu, C. *et al.* Structure-based inhibitor design for the intrinsically disordered protein c-Myc. *Sci. Rep.* **6**, 22298 (2016).

5) Page 8-9: Would it be possible to relate the observations in Fig. 2c/d to more structural characteristics of the c-Myc ensemble? The current interpretation is that two conformational ensemble intermediates are observed, but what do these ensembles correspond to (transiently formed alpha-helices, hydrophobic collapse, molten globule etc)? How can the observed temperature dependence be explained (taking into account thermodynamic considerations)?

Response: We thank the reviewer for the suggestions about the conformation structures and the temperature dependence. (i) According to the present experimental limitations of spatial and temporal resolution, currently it would be quite difficult to directly attribute specific conformational structures to current signals. In terms of the conductance levels and corresponding dwell times, we were able to explain that these current states were related to different sets of conformations with converging kinetic and thermodynamic properties. MD simulations have provided specific conformational structures of apo-c-Myc (*PLOS. Comput. Biol.* **2013**, *9*, e1003249; *Sci. Rep.* **2016**, *6*, 22298.). However, due to the limitation of the total simulation time, we were unable to match the simulation ensembles with the current states. We agree with the reviewer that the relationship between current states and specific conformational ensembles is very noteworthy to study, which require techniques with higher temporal resolution and more complex experimental designs. (ii) From a kinetic perspective, the dwell time of each current state decreased as the temperature increased, indicating that the molecular motion was accelerated with the increase of temperature. When the temperature reached 45 °C, a new current state with a longer dwell time emerged, related to the macroscopic Myc-Max binding properties. From a thermodynamic perspective, the instantaneous cluster signals in the blank buffer originated from the instantaneous conformational change of apo-LC46, which exhibited a nonequilibrium thermodynamics. Despite the long-time recording, there was no obvious regularity except for relative proportion according to the population distribution of different current states (**Fig. R2**). Therefore, we did provide further discussion in the revised main text.

Fig. R2 | Population distribution of different current states of apo-LC46. The data were analyzed from a current trajectory with cluster signals of the total time added up to 3 seconds.

6) Page 11: The authors determine the dissociation constant for the Myc-Max complex (Fig. 3d). Are these K_D values in agreement with literature reports?

Response: We thank the reviewer for the comment about the K_D . We performed the surface plasmon resonance (SPR) experiments to evaluate the K_D of biotinylated-Myc and Max peptides' binding. The results were shown in the **Fig. R3**. The K_D value was 351 ± 28 nM based on kinetic fitting, which is close to the K_D (205 ± 28 nM) derived from the single-molecule experiments.

Fig. R3 | Surface plasmon resonance assay of Myc with Max peptide. The association and dissociation curves of DQ47 at concentrations of 0.001–10 μM . The K_D value was 351 ± 28 nM based on kinetic fitting.

Our Revision: We have added **Fig. R3** as **Supplementary Fig. 30** in the Supplementary Information. And descriptions about the SPR experiments were also added on **Pages 14-15 in the Supplementary Information** as below:

SPR Experiments for biotinylated-Myc with Max peptides: SPR experiments were performed at 25 °C using a Biacore T200 (GE Healthcare Biacore, Uppsala, Sweden) instrument. Before the direct binding assays of DQ47, N-terminal biotinylated-Myc peptide (synthesized by GL Biochem Ltd.) was immobilized onto a SA chip (GE Healthcare) to a level of approximately 150 response units (RU). The experiments were performed in 1X PBS-P (GE Healthcare) with 5% DMSO. The certain concentration gradient DQ47 samples were applied over the surface at 30 $\mu\text{L}/\text{min}$ for 120 seconds association time with a dissociation time of 240 seconds. The raw data obtained were analysed with the Biacore T200 Evaluation Software 2.0 (GE Healthcare). The K_D values were calculated from the kinetic binding model fitting of the response-concentration plot to the equilibrium curves. The results were shown in Supplementary Fig. 30.

Page 12 in the main text: The K_D derived from single-molecule experiments was close to the SPR results ($K_D = 351 \pm 28$ nM, as shown in Supplementary Fig. 30).

7) It was not clear to which extent kinetic information can be extracted (k_{off} and k_{on} rates for the c-myc/max complex).

Response: We thank this reviewer for the comment. By using the QuB software (*Biophys. Rev. & Lett.* **2013**, 8, 191), the kinetic information can be extracted through the idealization of current trajectories of the LC46 (encounter state) association with DQ47 that allows stabilization of the fully folded LC46 in the heterodimer, as shown in Supplementary Fig. 32 and Supplementary Table 4 in the revised Supplementary Information. k_{bind} and k_{diss} correspond to k_{on} and k_{off} , respectively, which were derived from the reciprocal relation between rate constants and dwell times (**Eq. R1**).

$$k = \frac{1}{\tau} \quad (\text{Eq. R1})$$

Our Revision: We have revised the relevant statements **on Page 12 in the main text:** *The averaged dwell times (τ_{bind} and τ_{diss}) of bound and dissociated (encounter intermediate) states were derived through single exponential fitting of the dwell-time distributions from three different devices (Supplementary Figs. 31 and 32, Supplementary Table 4), further generating corresponding kinetic parameters, such as rate constants (k_{bind} and k_{diss} in Supplementary Table 4).*

Minor points:

- The legend to figure 1b should be completed to add explanations of the different subpanels.

Response: Thanks a lot for the suggestion on improving our manuscript. We have added a more detailed explanations of the subpanels in **the legend of Fig. 1b in the main text:**

The left panel shows the single-LC46-modified device in the dark field, taken under the excitation laser of 480 nm wavelength. The medium panel is the stochastic optical reconstruction of 5000 photos. The right panel is the magnification of the single reconstruction spot in the medium column.

- Please define the abbreviation “SiNW-FET” at the first place of use (not only in the abstract)

Response: Thanks for the suggestion on improving our manuscript. We have revised relevant descriptions on **Page 5 in the main text:**

Here, we built a c-Myc-modified molecular nanocircuit on a single-molecule electrical device platform with silicon nanowire field-effect transistors (SiNW-FETs) as biosensors (Fig. 1a)

- Figure 1c/d: It is not clear enough from the figure what is meant by “induction”. The color code is well-respected for disorder (yellow) and self-folding (green). Should the gray structure in panel d, actually be magenta to follow the logic in panel c?

Response: Thanks a lot for the detailed and constructive comment on Fig. 1. We apologize for our inaccurate description of the figure. The gray structure in panel d is the Max folded conformation. We have revised the Figure in the main text and make necessary annotations to the structure and current state.

Oure Revision: We have revised Fig. 1 as below.

Fig. 1 | Schematic demonstration and characterization of a c-Myc-modified SiNW-FET single-molecule device. *a*, Schematic diagram of the device architecture. *b*, Stochastic optical reconstruction microscopy characterization of c-Myc single-molecule modification. The left panel shows the single-LC46-modified device in the dark field, taken under the excitation laser of 480 nm wavelength. The medium panel is the stochastic optical reconstruction of 5000 photos. The right panel is the magnification of the single reconstruction spot in the medium column. *c*, Real-time

current trajectories (time resolution of 17 μ s) at different stages from left to right: unstructured (disordered) state of Myc (yellow), the transient self-folding process (green in the left panel), the Myc folding process with Max or Ligands (green in the right panel), and Myc–Max (ligands) binding equilibrium. d, Schematic diagram of c-Myc conformations at different interaction stages. The heterodimer structure was adapted from PDB 1NKP.

- What is the reason for using a DMSO-containing buffer for the experiments? Is that for comparability with the measurements using inhibitors? I am asking because some proteins may not be compatible with these conditions.

Response: Thanks a lot for the questions and comments. The DMSO was employed to increase the solubility of LC46. A DMSO-containing buffer was used in previous research on c-Myc and its inhibitors and it was shown that low concentration of DMSO had minor effect on the protein structure (*PLoS Comput. Biol.* **2013**, *9*, e1003249; *Sci. Rep.* **2016**, *6*, 22298).

- Please add temperature to the experiments in Fig. 2a.

Response: Thanks a lot for the suggestion. We have revised the legend of **Fig. 2a in the main text**:

Real-time current trajectory (time resolution of 17 μ s) of the c-Myc-modified device in a blank buffer solution at 25 °C.

We would like to take this opportunity to thank this reviewer for the precious time and important suggestions for us to improve the manuscript. We hope this reviewer will find this revised version satisfactory.

Sincerely,

The Authors

-----End of Response to Reviewer #1-----

Reviewer #2 (Remarks to the Author):

The manuscript describes the construction of a single-molecule electrical nanocircuit functionalised by the intrinsically disordered region of the cancer-associated protein c-Myc and its use to study conformational transitions in c-Myc either in its apo state or in response to binding to its binding partner Max as well as binding to a small molecule inhibitor. Overall, the manuscript is well written and the data figures are appropriate.

Major points:

1) A major conclusion of the manuscript relates to the potential identification of a pre-binding state. However, the interpretation of the "additional" experimental signal as a pre-binding state is questionable. In addition, even if this would be a pre-binding state the manuscript/data provide little insight into the nature of this pre-binding state

Response: We thank the reviewer for the constructive comment. We agree with the reviewer that the term of pre-binding state may bring ambiguity. This “additional” experimental signal may come from an intermediate ensemble formed by the folding of LC46 from the free state to the fully folded bound state with DQ47. It is more appropriate to refer this intermediate ensemble as encounter intermediate ensemble rather than pre-binding state. We have made necessary revisions in the revised manuscript.

Our Revision: We have revised the descriptions on the pre-binding state in the main text as below.

Page 2 in the main text: *Notably, apart from the unbound state, we captured a relatively stable encounter intermediate ensemble of Myc, a potential intermediate state suitable for drug targeting, under the transition path to certain fully bound state.*

Page 4 in the main text: *steady intermediate states of c-Myc to undergo conformational transition has not been distinguishable by conventional methods.*

Page 5 in the main text: *Different encounter intermediates were recognized through the competitive binding between inhibitors and DQ47, which further verified the interaction pathway.*

Page 11 in the main text: *Interestingly, the signals always went through a certain current state to finally form the binding complex, which was referred to as an encounter intermediate ensemble that bridges the labile conformation ensemble of free c-Myc to the fully folded state in the binding complex.*

Page 11 in the main text: *LC46 may first undergo a loose interaction system and fold to a dominant Myc*, which is favourable for further specific binding. This encounter*

intermediate ensemble was further enlarged along with the increase of DQ47 concentration, according to the increase of Myc population (5–100 nM DQ47 in Fig. 3c).*

Page 14 in the main text: *The encounter intermediate ensemble formed with 10074-A4 (Myc*¹) or I205 (Myc*²) could be derived from the above experiments (Figs. 4a and 4b).*

Page 14 in the main text: *The population of encounter intermediate ensembles declined and the LC46-inhibitor bound state enlarged as the inhibitor concentration increased.*

Page 14 in the main text: *...implying the presence of transient encounter intermediate and inhibitor-binding conformational ensembles even at this high inhibitor concentration.*

Page 14 in the main text: *The current signal pattern observed in the inhibitor concentration-dependent experiments further verified the encounter-intermediate pathway proposed in the above Myc-Max experiments.*

Page 15 in the main text: *With further increase of DQ47 concentration, the population of this new encounter intermediate ensemble increased until another bistate-like signal with a low conductance level appeared (Figs. 5b and 5d).*

Page 16 in the main text: *The decisive factor in this competition lies in the encounter intermediate ensembles of LC46 that is formed according to the microenvironment of the solution.*

Page 16 in the main text: *providing the possibility of specific encounter intermediate ensembles induced by inhibitors that were disfavoured for the heterodimerization of Myc with Max.*

Page 16 in the main text: *With changes of the relative concentration ratio of DQ47 and I205, LC46 tends to fold into different encounter intermediate ensembles, which further bind to either DQ47 or the inhibitor.*

Page 16 in the main text: *In addition, the competition experiments showed the presence of different encounter intermediate ensemble upon binding to different ligands.*

Page 17 in the main text: *An encounter intermediate mechanism for the Myc–Max interaction was also revealed by real-time electrical monitoring, including a folding transition among conformational intermediates at low Max concentrations and a binding equilibrium at high Max concentrations.*

Page 17 in the main text: *To the best of our knowledge, the encounter intermediate ensemble Myc* during the Max-binding process was captured experimentally for the first time. Furthermore, different encounter intermediate ensembles were observed...*

Page 17 in the main text: *...implying that encounter intermediate ensembles are the key to the function of the Myc IDR.*

Legend in Fig. 3a: blue for encounter intermediate ensemble, Myc*; and red for the fully folded binding complex.

2) The authors only focus on induced fit as a model for explaining their experimental data and completely neglect conformational selection, which is a bit surprising giving previous evidence of the presence of a transient α -helical region in the IDR of c-Myc prior to binding.

Response: Thanks a lot for this suggestive comment. We apologize for our negligence on the conformation selection mechanism. Relevant discussions have been added to the revised manuscript. We agree with the reviewer that the presence of transient folded conformation of LC46, which we also captured in our experiments. We calculated the k_{obs} as the number of occurrences per second of the fully folded binding complex. The k_{obs} at different DQ47 concentrations and different pHs is plotted in **Fig. R4**. The k_{obs} can be fitted by **Eq. R2** derived from a two-step induce-fit model (*Biochemistry* **2012**, *51*, 5894).

$$k_{obs} = k^{-} + k^{+} \frac{[L]}{K_D + [L]} \quad (\text{Eq. R2})$$

where k^{-} and k^{+} are the reverse and forward rate constant of the formation of a fully folded binding complex. $[L]$ can be approximate to the concentration in single-molecule experiments. K_D is the dissociation constant of the binding complex. However, this rectangular hyperbolic increase in k_{obs} with $[L]$ is not unequivocal evidence of the induced fit mechanism (*Biochemistry* **2012**, *51*, 5894; *Biophys. Chem.* **2014**, *189*, 33). As a result, we cannot conclude that LC46 either undergoes an induced-fit mechanism or a conformational selection mechanism. However, an intermediate state from free LC46 to the fully folded binding complex was detected. A relatively flat energy landscape was observed in the apo-LC46 experiments, where the most local minimums were unable to cause enough effective gating voltage. The presence of DQ47 and inhibitors gave rise to LC46 folding into several more stable conformational ensembles, which were captured by the SiNW-FET devices as the high frequent cluster signals. These partially folded ensembles tended to form a relatively stable one, which was favorable for the formation of the fully folded binding complex. The relative stable ensemble was regarded as encounter intermediate ensemble. Different encounter intermediate ensembles formed when LC46 interacted with different molecules. We have revised relevant descriptions about the mechanism in the revised manuscript.

Fig. R4 | k_{obs} at different DQ47 concentrations and different pHs. The k_{obs} is regarded as the number of occurrences per second of the fully folded binding complex. The k_{obs} can be fitted by Eq. R2 derived from a two-step induce-fit model. Red for pH 7.4 and dark red for pH 6.5.

Our Revision: We have revised the descriptions about the mechanism in the revised main text as below. **Eq. R2** and **Fig. R4** are also added as **Supplementary Eq. 2** and **Supplementary Fig. 37** in the Supplementary Information.

Page 4 in the main text: *Although several models have been proposed to explain the interactions between IDPs and their partners, such as coupled folding¹², binding-induced-folding³⁷, and conformational selection³⁸, steady intermediate states of c-Myc to undergo conformational transition has not been distinguishable by conventional methods³⁶.*

[12] Sugase, K., Dyson, H. J. & Wright, P. E. Mechanism of coupled folding and binding of an intrinsically disordered protein. *Nature* **447**, 1021-1025 (2007).

[37] Turjanski, A. G., Gutkind, J. S., Best, R. B. & Hummer, G. Binding-induced folding of a natively unstructured transcription factor. *PLoS Comput. Biol.* **4**, e1000060 (2008).

[38] Tsai, C. J., Ma, B., Sham, Y. Y., Kumar, S. & Nussinov, R. Structured disorder and conformational selection. *Proteins* **44**, 418-427 (2001).

Page 10 in the main text: *According to the chronological order and the alternating nature of the current signals, we assumed that the high-frequency clustered current signals represented the folding process of LC46 with multiple labile intermediate conformation ensembles. The bistate signals with longer dwell times originated from the dimerization process of LC46 and DQ47. The lower current state in the bistate signals corresponded to the fully folded binding complex, which was much more stable than the partially folded conformational ensembles. Because of the global positive charge of DQ47 and LC46 as well as the compact HLH-zip structure, a significant change of the effective gating voltage occurred, leading to a lower current level (Figs.*

3b and 3c). The gating voltage was mainly affected by charge distribution changes induced by the dramatic conformational changes. Interestingly, the signals always went through a certain current state to finally form the binding complex, which was referred to as an encounter intermediate ensemble that bridges the labile conformation ensemble of free c-Myc to the fully folded state in the binding complex.

Page 11 in the main text: Based on these concentration-dependent signal patterns, we propose an interaction hypothesis (Fig. 1d) for LC46 and DQ47. At a very low DQ47 concentration (5 nM), LC46 may first undergo a loose interaction system and fold to a dominant Myc*, which is favourable for further specific binding. This encounter intermediate ensemble was further enlarged along with the increase of DQ47 concentration, according to the increase of Myc* population (5–100 nM DQ47 in Fig. 3c). As DQ47 concentration further increase, the equilibrium of the binding process would finally shift to the fully folded binding structure (100 nM – 5 μ M DQ47 in Fig 3c).

Page 13 in the main text: The fast-clustered pulse signals with higher conductance levels should result from the partial folding of LC46 in the presence of inhibitor. In comparison with the Myc* state formed with DQ47, the dominant bistate binding signals were observed at a much higher concentration of inhibitors due to their weak binding strength, and the pulsed spike current signals remained.

Page 14 in the main text: To explore the interaction process between LC46 and small molecule inhibitors, concentration-dependent experiments were performed (Supplementary Figs. 33 and 34).

Page 14 in the main text: The current signal pattern observed in the inhibitor concentration-dependent experiments further verified the encounter-intermediate pathway proposed in the above Myc-Max experiments.

Page 16 in the main text: In addition, the competition experiments showed the presence of different encounter intermediate ensemble upon binding to different ligands, and again demonstrated the encounter-intermediate pathway for the Myc-Max interaction.

Page 17 in the main text: An encounter intermediate mechanism for the Myc-Max interaction was also revealed by real-time electrical monitoring, including a folding transition among conformational intermediates at low Max concentrations and a binding equilibrium at high Max concentrations.

Page 17 in the main text: The k_{obs} (the number of binding complex states per second) was plotted versus Max concentration to investigate the mechanism of Myc-Max (Supplementary Fig. 37). Although the k_{obs} could be well fitted by a two-step induce-fit model⁵⁹, it remained unsure whether LC46 underwent an induced-fit mechanism or a conformational selection mechanism^{59,60}.

[59] Vogt, A. D. & Di Cera, E. Conformational selection or induced fit? A critical appraisal of the kinetic mechanism. *Biochemistry* **51**, 5894-5902 (2012).

[60] Gianni, S., Dogan, J. & Jemth, P. Distinguishing induced fit from conformational selection. *Biophys. Chem.* **189**, 33-39 (2014).

We would like to take this opportunity to thank this reviewer very much for his/her precious time and important suggestions to improve the manuscript. We hope this reviewer will find this revised version satisfactory.

Sincerely,

The Authors

-----End of Reply to Reviewer #2-----

Reviewer #3 (Remarks to the Author):

This manuscript by Guo, Lai, and co-workers describes exceptionally creative and ground-breaking single molecule studies. Intrinsically disordered proteins (IDPs) would appear to be poor subjects for single molecule experiments, as random conformational changes would frustrate analysis. However, perhaps through clever choice of a target subject, the authors may have demonstrated the repeatable motions of an IDP in a number of interesting states, including control, binding to ligand, and binding to ligand in competition with an inhibitor. For the following reasons, however, this reviewer is not convinced that the authors are observing what they claim.

1. The agreement in the HRMS characterization between theoretical and observed MWs is suspiciously high. Such agreement might happen 5% of the time, but every molecule with a tiny difference in the submilli-amu seems too good to be true. I do not believe the authors have made the molecules they claim to have made and suggest that they use the checklist for characterization required for organic chemicals by organic chemistry journals (e.g., JOC). In other words, I'd like to see a fully assigned ^1H and ^{13}C NMR spectra and LC-MS to check purity, etc. Without such data, the reader cannot believe the results as presented.

Response: We thank this reviewer for the comment and suggestion. The two inhibitors used in this study were not synthesized by ourselves, but purchased from TopScience Co., Shanghai, China (10074-A4) and Shenzhen Biochemilogic Technology Co. Ltd. (PKUMDL-YC-1205). Both compounds were commercially available. The companies provided NMR, LC-MS, or both. Here, we rechecked the purity of PKUMDL-YC-1205 and 10074-A4 by performing HPLC, MS, ^1H and ^{13}C NMR. Please see the fully assigned NMR, MS and HPLC spectra of the inhibitors used in the study (**Figs. R5-R12**). The purity was more than 90% for both compounds. Please note that both compounds have been used in previously published studies (*Sci. Rep.* **2016**, *6*, 22298.). Also, we provided the HPLC and MS spectra of LC46 and DQ47 provided by the supplier (GL Biochem. Ltd., Shanghai, China, **Figs. R13-R16**). Unlike the inhibitor molecules, HPLC and MS spectra are sufficient for the routine identification of peptides.

Fig. R5 | MS spectrum of 10074-A4.

Fig. R6 | ¹H NMR of 10074-A4 in DMSO-*d*₆.

Fig. R7 | ^{13}C NMR of 10074-A4 in $\text{DMSO-}d_6$.

Signal: DAD1 A, Sig=254,4 Ref=360,100

RT [min]	Type	Width [min]	Area	Height	Area%	Name
6.953	BV	0.1112	229.3759	32.1635	0.8666	
8.126	MM	0.1899	24050.2598	2110.7236	90.8604	
8.722	BV	0.0958	2026.5587	329.1034	7.6562	
10.184	VV	0.0900	163.2565	27.9897	0.6168	
Sum			26469.4509			

Fig. R8 | HPLC spectrum of 10074-A4.

Fig. R9 | MS spectrum of PKUMDL-YC-1205.

Fig. R10 | ¹H NMR of PKUMDL-YC-1205 in DMSO-d₆.

Fig. R11 | ^{13}C NMR of 1205 in DMSO- d_6 .

Signal: DAD1 A, Sig=254,4 Ref=360,100

RT [min]	Type	Width [min]	Area	Height	Area%	Name
2.059	BV	0.1283	77.9492	9.0674	0.2370	
2.212	VB	0.2151	151.0297	8.7618	0.4593	
3.262	BB	0.1654	28.1552	2.4617	0.0856	
5.620	BV	0.0911	50.6491	8.8125	0.1540	
7.083	BB	0.1209	88.3691	10.8587	0.2687	
8.766	BV	0.1858	32405.7129	2557.4395	98.5453	
9.133	VV	0.0995	82.2197	12.0686	0.2500	
Sum			32884.0848			

Fig. R12 | HPLC spectrum of PKUMDL-YC-1205.

Fig. R13 | MS spectrum of LC46 provided by the supplier.

Fig. R14 | HPLC spectrum of LC46 provided by the supplier.

Fig. R15 | MS spectrum of DQ47 provided by the supplier.

Fig. R16 | HPLC spectrum of DQ47 provided by the supplier.

Our Revision: We have added the fully assigned spectra of NMR, MS and HPLC, **Figs. R5–R12 as Supplementary Figs. 5–12** in the revised **Supplementary Information**. The HPLC and MS spectra of LC46 and DQ47 provided by the supplier, **Figs. R13–R16**, were also added as **Supplementary Figs 1–4** in the revised **Supplementary Information**. In addition, we have added relevant description and make some revision of the multiplet analysis on **in the Supplementary Information**:

Page 4 in the Supplementary Information: Samples of LC46 and DQ47 were synthesized by GL Biochem. Ltd., Shanghai, China with purity of more than 95%, which was confirmed by the supplier, using HPLC and MS (Supplementary Figs. 1-4).

Page 4 in the Supplementary Information: The company provided NMR and LC-MS. Here, we rechecked the purity of PKUMDL-YC-1205 and 10074-A4 by performing HPLC, MS, ^1H and ^{13}C NMR. Both compounds are over 90% pure, which was determined by an Agilent 1206 Infinity high performance liquid chromatography instrument with a SB-C18 column (4.6 mm \times 150 mm, 5 μm) with methanol and water with 0.1% formic acid as the mobile phase. The flow rate was 1 mL/min and the peak was detected at 254 nm. ^1H NMR spectra were recorded on a Bruker 500 MHz spectrometer. ^{13}C NMR spectra were recorded on a Bruker 400 MHz spectrometer. The chemical shift values (δ) are reported in ppm relative to tetramethylsilane as the internal standard. ^1H spectra were represented as follows: chemical shift, multiplicity (s = singlet, d = doublet, t = triplet, m = multiplet), coupling constant (J values) in Hz and integration. ^{13}C NMR spectra were represented as follows: chemical shift, multiplicity (d = doublet, m = multiplet), coupling constant (J values) in Hz.

Page 5 in the Supplementary Information: The fully assigned spectra of NMR, MS and HPLC have been provided as below (Supplementary Figs. 5-12).

Page 5 in the Supplementary Information: Compound 10074-A4: ^1H NMR (500 MHz, $\text{DMSO-}d_6$) δ 8.32 (t , J = 1.7 Hz, 2H), 7.67 (d , J = 8.8 Hz, 2H), 7.49 (dt , J = 8.8, 1.6 Hz, 2H), 5.28 (dd , J = 5.6, 1.7 Hz, 1H), 4.47 (dd , J = 15.1, 4.0 Hz, 1H), 4.35 (dd , J = 15.0, 8.0 Hz, 1H), 4.22–4.16 (m , 1H), 4.14 (s , 2H), 3.73 (ddd , J = 13.5, 8.8, 1.5 Hz, 1H), 3.54 (dd , J = 13.6, 3.9 Hz, 1H). ^{13}C NMR (101 MHz, $\text{DMSO-}d_6$) δ 172.44, 172.12, 139.49, 126.09, 123.56, 122.48, 120.25, 111.76, 66.04, 46.96, 45.41, 33.81.

Page 5 in the Supplementary Information: Compound PKUMDL-YC-1205: ^1H NMR (500 MHz, $\text{DMSO-}d_6$) δ 12.70 (s , 1H), 8.60 (s , 1H), 7.90 (d , J = 7.6 Hz, 2H), 7.72 (d , J = 7.5 Hz, 2H), 7.59 (d , J = 8.3 Hz, 1H), 7.42 (q , J = 7.1 Hz, 2H), 7.31 (dd , J = 14.5, 7.4 Hz, 2H), 7.27–7.14 (m , 15H), 4.36 (dd , J = 10.3, 6.9 Hz, 1H) 4.32–4.20 (m , 3H), 2.67 (d , J = 7.1 Hz, 2H). ^{13}C NMR (101 MHz, $\text{DMSO-}d_6$) δ 173.25, 168.77, 155.82, 144.71, 143.79, 140.73, 128.58, 127.86 – 126.88 (m), 126.37, 125.22 (d , J = 4.2 Hz), 120.14, 69.45, 65.75, 51.06, 46.66, 38.04.

Page 6 in the main text: The materials and detailed methods are provided in the Supplementary Information (Supplementary Figs. 1–16).

2. The authors claim that only one protein is attached per SiNW-FET. However, they show this using a FITC labeling of the protein and then imaging to observe only one label. First, FITC labelling is not quantitative. The conditions could label a low percentage of proteins that are present. Second, FITC is readily photo-bleached, which

again would lower the yield of active FITC labels present. The authors could use in-liquid AFM to rigorously demonstrate single molecule attachment. Or they could moderate the manuscript's claims.

Response: Thanks for the suggestion. We agree with this reviewer on the unreliability of FITC labels because of the un-quantified factors and the low yield of active FITCs. In order to further validate the single-LC46 modification, we conducted AFM experiments (**Fig. R1**) to verify the single-LC46 modification.

Our Revision: We have added **Fig. R1** as **Supplementary Fig. 18** in the **Supplementary Information**. Relevant descriptions about AFM image have been added **on Page 6 in the main text**:

The device without FITC modification was also characterized using AFM (Supplementary Fig. 18), verifying that the device was successfully modified with a single LC46 molecule.

More detailed descriptions about the AFM and optical characterization have been added **on Page 14 in the revised Supplementary Information** as below:

The device without FITC modification was also characterized using atomic force microscopy (AFM, Bruker AFM Dimension Icon). The AFM image (Supplementary Fig. 18) was generated at the ScanAsyst mode with a sampling rate of 1.00 Hz and 512 samples per line, which confirmed the presence of a single LC46 attached to the etched gap on the side of silicon nanowires.

3. To be blunt, the data looks too clean for single molecule data. I've looked at a lot (a lot!) of single molecule data in my lab and others. The histograms do not look so tight and there's considerably more spread in the signals observed. For an IDP, I'd expect even more spread, not less, as one could imagine a number of variant conformations accessed by a less tightly defined structure. Has this data been subjected to anti-aliasing or other signal processing to force it into two or three levels?

Response: Thanks a lot for the comments and suggestions. (i) We used a lock-in amplifier and a low-pass Butterworth filter to reduce the high-frequency signals to obtain a higher signal to noise ratio. (ii) We agree with the reviewer's opinion that there are indeed many different states observed. The peaks in the current distribution histograms can be further split according to the different current levels. The same current level can also be separated in terms of different dwell times. However, the cluster signals arised from the conformational change of LC46 exhibited a nonequilibrium thermodynamics. As a result, despite the long-time recording, whether

the wide-spread peaks can be further split into different states still requires more information. Further detailed analysis can be challenging and need technologies with higher temporal resolution and multiple dimensions. We are now making effort in this direction.

4. Would the authors also please discuss the time resolution of their experiments? I believe the “instantaneous” rise times are not due to instant transitions, but due to the lower time resolution of their experiments than the time required to move between conformational states. However, it would be important to clarify this point for the readers.

Response: Thanks a lot for the constructive suggestions. We agree with the reviewer’s opinion on the relatively lower time resolution (17 μ s) in contrast with the time scale of conformational change (\sim 1 μ s). We sincerely apologize for the inaccurate expressions in our manuscript. The temporal resolution of our experiments was mentioned on Pages 7–8 in the main text. As a result, we considered that the changes between the current states represented the changes between conformational ensembles rather than conformational structures.

Our Revision: The descriptions of the time resolution have been added to the legend of each *i-t* figure in the main text, and the expression of “instantaneous” has been clarified in the main text.

Pages 7–8 in the main text: *According to the time resolution of 17 μ s in the experiments, the instantaneous signal rise and decrease between the current states corresponded to the changes between conformational ensembles rather than conformational changes (\sim 1–10 μ s)⁵⁶.*

[56] Yu, H., Siewny, M. G. W., Edwards, D. T., Sanders, A. W. & Perkins, T. T. Hidden dynamics in the unfolding of individual bacteriorhodopsin proteins. *Science* **355**, 945-949 (2017).

Legend to Fig. 1c: *Real-time current trajectories (time resolution of 17 μ s) at different stages of the Myc–Max interaction...*

Legend to Fig. 2a: *Real-time current trajectory (time resolution of 17 μ s) of the c-Myc-modified device in a blank buffer solution...*

Legend to Fig. 3a: *Real-time trajectory over 60 s of the Myc–Max interaction process and a histogram of the current data (time resolution of 17 μ s).*

Legend to Figs. 4a and b: *Real-time current trajectories (time resolution of 17 μ s)...*

Legend to Figs. 5a–d: *Real-time current trajectories (time resolution of 17 μ s)...*

Minor issues:

1. What are the dimensions of the SiNW-FET? What is the composition of the PBS used (PBS can vary somewhat amongst different investigators)?

Response: The SiNWs were grown with a diameter of 30–40 nm and a length over 10 μm . The 1 \times PBS buffer was ordered from M&C Gene Technology (Catalogue#: CC008), which contains 8g/L NaCl, 0.2 g/L KCl, 1.44 g/L Na₂HPO₄ and 0.24 g/L KH₂PO₄ at pH 7.4.

Our Revision: The information of the PBS buffer has been added on **Page 4 in the Supplementary Information:** *The 1 \times PBS buffer was ordered from M&C Gene Technology (Catalogue number: CC008), which contains 8g/L NaCl, 0.2 g/L KCl, 1.44 g/L Na₂HPO₄ and 0.24 g/L KH₂PO₄ at pH 7.4.*

p. 3 of SI: catalog is misspelled.

Our Revision: We have corrected the misspelling on **Page 4 in the revised Supplementary Information:**...*(Catalogue number: STK834743)*...

We would like to take this opportunity to thank this reviewer very much for all the time involved and important suggestions for us to improve the manuscript. We hope this reviewer will find this revised version satisfactory.

Sincerely,

The Authors

-----End of Response to Reviewer #3-----

Reviewer #4 (Remarks to the Author):

In this manuscript, the authors reported dynamic interactions between an intrinsically disordered protein, c-Myc and its binding partners (Max/inhibitors) using a single-molecule nanocircuits approach, where the single protein is attached to the silicon nanowire field-effect transistor (SiNW-FET) via linker molecules. The authors observed current fluctuations between multiple current levels during dose-dependent measurements and claimed that those current levels correspond to Myc's intermediate conformational states including disordered state, partially folded state, fully folded state, and bound state with either Max or inhibitors. In addition, the authors showed temperature dependence of a few conformational states, which could be responsible for the self-folding process. However, the conclusions drawn from the results and discussion reported in this manuscript are based solely on the simple assignment of current fluctuations to potential conformational states of the protein without any direct and/or supporting validation (e.g., crystal structure, bulk assay, alternative single molecule measurements such as FRET or EPR (DEER)). In addition, this reviewer finds that the manuscript lacks new information or insight into the mechanisms of protein folding and protein-protein binding. Further details are outlined below.

1. What would be the difference between the binding-induced-folding described here and the induced-fit model?

Response: Thanks for the suggestive comment. We think that binding-induced-folding is the same as the induced-fit model for IDPs (*Curr. Opin. Struct. Biol.* **2009**, *19*, 31). However, after conscientious consideration on the mechanism, we cannot be sure that LC46 undergoes an induced-fit mechanism or a conformational selection mechanism. Therefore, we focus our study more on the conformation intermediate ensemble regarded as encounter intermediate ensemble in our revised manuscript.

Our Revision: We have revised the descriptions about the mechanism in the main text as below.

Page 10 in the main text: *According to the chronological order and the alternating nature of the current signals, we assumed that the high-frequency clustered current signals represented the folding process of LC46 with multiple labile intermediate conformation ensembles. The bistate signals with longer dwell times originated from the dimerization process of LC46 and DQ47. The lower current state in the bistate signals corresponded to the fully folded binding complex, which was much more stable than the partially folded conformational ensembles. Because of the global positive*

charge of DQ47 and LC46 as well as the compact HLH-zip structure, a significant change of the effective gating voltage occurred, leading to a lower current level (Figs. 3b and 3c). The gating voltage was mainly affected by charge distribution changes induced by the dramatic conformational changes. Interestingly, the signals always went through a certain current state to finally form the binding complex, which was referred to as an encounter intermediate ensemble that bridges the labile conformation ensemble of free c-Myc to the fully folded state in the binding complex.

Page 11 in the main text: *Based on the concentration-dependent signal patterns, we propose an interaction hypothesis (Fig. 1d) for LC46 and DQ47. At a very low DQ47 concentration (5 nM), LC46 may first undergo a loose interaction system and fold to a dominant Myc*, which is favourable for further specific binding. This encounter intermediate ensemble was further enlarged along with the increase of DQ47 concentration, according to the increase of Myc* population (5–100 nM DQ47 in Fig. 3c).*

Page 12 in the main text: *the higher current state was the encounter conformational intermediate Myc* formed by the transient release of DQ47 from the heterodimer.*

Page 13 in the main text: *The fast-clustered pulse signals with higher conductance levels should result from the partial folding of LC46 in the presence of inhibitor. In comparison with the Myc* state formed with DQ47, the dominant bistate binding signals were observed at a much higher concentration of inhibitors due to their weak binding strength, and the pulsed spike current signals remained.*

Page 14 in the main text: *To explore the interaction process between LC46 and small molecule inhibitors, concentration-dependent experiments were performed (Supplementary Figs. 33 and 34).*

Page 14 in the main text: *The current signal pattern observed in the inhibitor concentration-dependent experiments further verified the interaction process proposed in the Myc-Max experiments.*

Page 16 in the main text: *In addition, the competition experiments showed the presence of different encounter intermediate ensembles upon binding to different ligands, and again demonstrated the encounter-intermediate process for the Myc-Max interaction.*

Page 17 in the main text: *An encounter intermediate mechanism for the Myc-Max interaction was also revealed by real-time electrical monitoring, including a folding transition among conformational intermediates at low Max concentrations and a binding equilibrium at high Max concentrations.*

2. The device schematic image (Fig 1a) is not scaled, which may confuse readers. The authors need to provide more information on the dimension of the nanogap and the

scale of LC46? If the nanogap is too large, DQ47/inhibitors could interact directly with SiNW, regardless of the presence or absence of LC46, leading to similar random telegraph signals. If the nanogap is tight, the LC46 would be confined in the nanogap, limiting its conformational motions and increasing its potential to be sticking to the SiNW or Su-8 if it is in a disordered, long chain-like state as depicted in Fig1a.

Response: Thanks a lot for the comments and suggestions. The information on the dimension of the nanogap and the scale of LC46 was shown in **Fig. R17**. We agree with the reviewer's opinion about the nanogap. According to the AFM image, the gap size is about 10-nm deep and about 15-nm wide. It is difficult to accurately control the gap size, which may have certain influence on the molecule motion. Further development of a gap-size control method is needed. However, the influence of the gap size on the binding is relatively small according to control experiments using molecule-linkage devices without LC46 modification in the DQ47 and inhibitors solutions (**Fig. R18**).

Fig. R17 | Dimension information of a LC46 modified device. Prediction of the molecular length of apo-LC46 and molecule linkage.

Fig. R18 | Control experiments using molecule-linkage devices without LC46 modification. a–c, Real-time current trajectories of a molecule-linkage modified device in different solutions (PBS buffer [8g/L NaCl, 0.2 g/L KCl, 1.44 g/L Na₂HPO₄ and 0.24 g/L KH₂PO₄], pH = 7.4, 5% DMSO): 100 nM DQ47 (a), 6.25 μM 10074-A4 (b), 6.25 μM PKUMDL-YC-1205 (c). The right column is the 60 s current data, and the left column is the magnified view of the selected area.

Our Revision: We have added **Fig. R17** as **Supplementary Fig. 21** and **Fig. R18** as **Supplementary Fig. 26** and in the Supplementary Information. Relevant descriptions have been added in the main text as below.

Page 7 in the main text: *A 0.01 × PBS solution was used to acquire appropriate Debye length and enough signal-to-noise ratio according to the PBS concentration dependent experiments (Supplementary Figs. 20 and 21, Supplementary Table 1).*

Pages 9-10 in the main text: *Control experiments showed no obvious signals when a device without LC46 was measured in DQ47 solution, indicating that signals in Fig. 3a originated from the interaction between LC46 and DQ47 (Supplementary Fig. 26a).*

Page 13 in the main text: *Control experiments indicated that the signals in Fig. 4 came from the interaction between LC46 and inhibitors (Supplementary Figs. 26b and 26c).*

3. The authors need to explain the method used to control the attachment of a single protein. What was the yield? Also, was the attachment stable for the long-duration measurement? There could be dissociation, significant degradation, unfolding or

misfolding during the long-term, temperature-dependent measurements. It is important to know how many devices and proteins were tested in the manuscript, and confirm that the signals are reproducible and the dwell times are with different binding partners and temperatures.

Response: Thanks a lot for the suggestions. The yield of the single-LC46 devices was $\sim 10\%$ on each chip. For each experiment, we at least tested three different devices to verify that the signals analyzed were reproducible. We agree with the reviewer's opinion about the stability of the attachment. The temperature and long-time measurement did have an impact on the device. When temperature reached above $45\text{ }^{\circ}\text{C}$, the device experienced irreversible changes and the signals could not be detected. However, the stability of the device can at least ensure us to obtain the data we needed. The signals were reproducible according to **Fig. R19**. The dwell times were different at different temperature but similar when different devices were measured under the same condition.

Fig. R19 | Experiments measuring different devices in 1 μ M DQ47 solution at 37 $^{\circ}$ C. a–c, Real-time current trajectories from different devices in 1 μ M DQ47 solution at 37 $^{\circ}$ C (0.01 \times PBS [8g/L NaCl, 0.2 g/L KCl, 1.44 g/L Na₂HPO₄ and 0.24 g/L KH₂PO₄], pH = 7.4, 5% DMSO): Device 1 (a), Device 2 (b), and Device 3 (c). The right column is the 100 s current data, and the left column is the magnified view of the selected area. d–f, Dwell-time distributions of the Myc-Max binding process from different devices: Device 1 (d), Device 2 (e), and Device 3 (f).

Our Revision: We have added **Fig. R19** as **Supplementary Fig. 31**. The information about the yield and the stability of devices, reproducible signals has been added in the **Supplementary Information**.

Page 14 in the Supplementary Information: *The yield of the single-LC46 devices was ~10% on each chip. The stability of the devices can ensure us to obtain the data we needed (~8–12 h). When temperature reached above 45 $^{\circ}$ C, the device experienced irreversible changes. The conductance of the devices might change dramatically with large difference values, after which signals could not be detected.*

Page 20 in the Supplementary Information: *For statistical analysis, we at least tested three different devices to verify that the signals analyzed were reproducible (as shown in Supplementary Fig. 31).*

Page 12 in the main text: *The averaged dwell times (τ_{bind} and τ_{diss}) of bound and dissociated (encounter intermediate) states were derived through single exponential fitting of the dwell-time distributions from three different devices (Supplementary Fig. 31 and 32, Supplementary Table 4).*

4. There are no details on stochastic optical reconstruction (microscopy) measurements performed on the Silicon devices, which are crucial to demonstrate a single protein attachment.

Response: Thanks a lot for the comment. In order to verify the single-LC46 functionalization with fluorescence methods, we introduced FITC as the fluorescent group. Since there are multiple lysine reactive sites in the LC46 sequence, we carried out the reaction at a low FITC concentration to reduce the reactivity. We utilize stochastic optical reconstruction microscopy (STORM) to realize fluorescent characterization at the single-molecule scale. In order to further validate this conclusion, we conducted AFM experiments (**Fig. R1**) to verify the single-LC46 modification.

Our Revision: We have added **Fig. R1** as **Supplementary Fig. 18** in the **Supplementary Information**. Relevant descriptions about STORM and AFM image

have been added on **Page 6 in the main text**:

Fluorescence images and stochastic reconstruction results showed a single location spot with a resolution of ~20 nm, corresponding to a single fluorescence molecule (Fig. 1b and Supplementary Fig. 17). The device without FITC modification was also characterized using atomic force microscopy (Supplementary Fig. 18), verifying that the device was successfully modified with a single LC46 molecule.

More detailed descriptions about the AFM and optical characterization have been added on **Page 14 in the revised Supplementary Information** as below:

A PBS buffer liquid layer was added to the surface of the LC46-modified SiNW FET device firstly. The device was then covered by a coverslip and placed on the microscope objective stage. A Nikon Ni-E microscope with a $\times 100$ objective lens was positioned in close contact with the coverslip on the device through the lens oil. The field of view of the stochastic optical reconstruction microscopy (STORM, N-STORM Nikon) was zoomed on the modified device. Then, the FITC-modified molecules on the devices were excited by a laser of 480 nm, and an EMCCD (Andor) was used to record the emitted signals with a 50-ms exposure resolution. The switching fluorescent signals of the target location were recorded within 5 min. Subsequently, the reconstruction and analysis of the optical images were carried out with the Advanced Research software⁴. Only a single fluorescence spot was reconstructed on individual SiNW-FET devices (Fig. 1b and Supplementary Fig. 17). The device without FITC modification was also characterized using atomic force microscopy (AFM, Bruker AFM Dimension Icon). The AFM image (Supplementary Fig. 18) was generated at the ScanAsyst mode with a sampling rate of 1.00 Hz and 512 samples per line, which confirmed the presence of a single LC46 attached to the etched gap on the side of silicon nanowires.

5. The electronic characteristics of SiNW FETs, such as the IV and IVg, should be presented here so that the readers can assess the gating effects of small protein charges.

Response: Thanks a lot for the suggestion. The electronic characteristics of SiNW-FETs were shown in **Fig. R20** as below. Bottom gate and liquid gate were both used to measure the electrical characteristics of SiNW-FETs.

Fig. R20 | Electrical characterization of SiNW-FETs. a, Output curves of a *p*-type SiNW-FET. V_{SD} scanned from 0 to 0.5 V at a certain V_G (from -15 to 15 V with an interval of 3 V). **b,** Transfer curve of a *p*-type SiNW-FET. The device was measured under liquid gate by using a platinum probe as the third electrode in a PBS buffer solution (red line: from -0.3 to 0.3 V, black line: from 0.3 to -0.3 V, grey line: gate leakage current).

Our Revision: The detailed descriptions of the electrical characterization have been added on **Page 7 in the Supplementary Information** as below. **Fig. R20** has been also added as **Supplementary Fig. 15** in the Supplementary Information.

Page 12 in the Supplementary Information: *The boron-doped Si substrate was employed as the global bottom gate with a 1000 nm SiO₂ layer as the dielectric layer. SiNW-FET devices were located on a Karl Süss (PM5) manual probe station, and the electrical characterization was then conducted at room temperature using an Agilent 4155 C semiconductor analyzer. SiNW-FETs showed the typical *p*-type behaviors with good Ohmic contacts (Supplementary Fig. 15). A platinum probe was also used as a liquid gate electrode with a droplet of PBS buffer solution on the device as the dielectric layer.*

Page 6 in the main text: *The materials and detailed methods are provided in the Supplementary Information (Supplementary Figs. 1–16).*

6. (Page7, line 10). The authors need to specify the positive charges on the surface of LC46 and their location relative to the SiNW, and if they are within the Debye length. What is the Debye length in this work? How many charges contribute to signal generation? Also, how much protein charges can induce or modulate entire NW conductivity? The author could test different pH, buffers, and ionic strengths to validate signal generation. Also, the authors could repeat these experiments with selective

mutation of LD46 (different charges) to verify the signal direction and amplitude.

Response: Thanks a lot for the suggestions. As shown in Fig. R1, the length of LC46 is about 5.3 nm with a 1.7-nm long linkage, added up to ~7 nm from the surface of SiNWs. To fully detected the conformational change of the LC46 and its association with other ligands, PBS concentration-dependent experiments were conducted to explore the effect of ionic strength. The results are shown in Fig. R21. In contrast of the 1× PBS solution without obvious signals, we captured changes of the current level in 0.1× PBS and 0.01× PBS solutions. The signals showed more details in 0.01× PBS because of the longer Debye length. The Debye lengths of different PBS concentrations can be calculated by Eq. R3 as below.

$$\lambda_D = \sqrt{\frac{\epsilon k_B T}{q^2 c}} \quad (\text{Eq. R3})$$

where ϵ is the dielectric permittivity of the media, k_B is Boltzmann's constant, T is the temperature, q is the electron charge, and c represents the ionic strength of the electrolyte (*Nano Lett.* **2012**, *12*, 5245.). **Table R1** shows the Debye lengths of solutions in our experiments.

In addition, after the mutation, the properties of disordered peptide will change, further bringing about more complex changes, which is not the current focus of this work and can be further explored in the future.

Fig. R21 | PBS concentration-dependent experiments. a–c, Real-time current trajectories of a single-Myc modified device in different PBS solutions (PBS buffer

[8g/L NaCl, 0.2 g/L KCl, 1.44 g/L Na₂HPO₄ and 0.24 g/L KH₂PO₄], pH = 7.4, 5% DMSO): 1× PBS (a), 0.1× PBS (b), 0.01× PBS (c). The right column is the 60-s current data, and the left column is the magnified view of the selected area.

Table R1. Debye lengths of different PBS concentrations.

PBS pH 7.4 (5% DMSO)	λ_D (nm)	ionic strength
×1	0.72	143 mM
×0.1	2.4	14 mM
×0.01	7.5	1.4 mM

Our Revision: We have added **Fig. R21** as **Supplementary Fig. 20**, **Eq. R3** as **Supplementary Eq. 3** and **Table R1** as **Supplementary Table 1** in the Supplementary Information. We also added relevant description in the main text as below. The descriptions of the buffer in Supplementary Figures are omitted here but can be found in the legend of each figure containing real time data in the **revised Supplementary Information**.

Pages 6–7 in the main text: *In control experiments, bare and maleimide-modified devices were monitored in a blank condition (phosphate-buffered saline [PBS, ×0.01, 8g/L NaCl, 0.2 g/L KCl, 1.44 g/L Na₂HPO₄ and 0.24 g/L KH₂PO₄], 5% dimethylsulfoxide [DMSO], pH = 7.4).*

Page 7 in the main text: *A 0.01× PBS solution was used to acquire the appropriate Debye length and enough signal-to-noise ratio according to the PBS concentration dependent experiments (Supplementary Figs. 20 and 21, Supplementary Table 1).*

Page 9 in the main text: *sophisticated multi-state current signals were observed during real-time monitoring (0.01× PBS with DQ47, 5% DMSO, pH = 7.4) (Fig. 3a, Supplementary Figs. 25a–25e).*

Legend to Fig. 2a: *(phosphate-buffered saline [PBS, ×0.01, 8g/L NaCl, 0.2 g/L KCl, 1.44 g/L Na₂HPO₄ and 0.24 g/L KH₂PO₄], 5% dimethylsulfoxide [DMSO], pH = 7.4).*

Legend to Fig. 4a and b: *(0.01× PBS, 5% DMSO, pH = 7.4).*

Legend to Fig. 5a and b: *(0.01× PBS, pH = 7.4, 100 μM 10074-A4, and 100 nM DQ47).*

Legend to Fig. 5c and d: *(0.01× PBS buffer, pH = 7.4, 100 μM 1205, and 100 nM DQ47)*

7. The electronic signals presented in all figures appear to be simple 1-dimensional motions of the protein, however, if LD46 can move and rotate freely, its conformations would not remain consistent. Thus, the gating effects of the protein would change during the conformational transition.

Response: Thanks a lot for the comment. We think that the electric signals originated

not only from 1-dimensional motions of LC46, but from the changes of total effective gating caused by the global conformational changes of LC46. The changes of LC46 with total positive charge from a relatively extending conformation ensemble to a partially folded or compact binding conformation will lead to a change of the effective gating, therefore giving rise to the decrease of currents.

8. The authors need to provide more data or compelling arguments for the transient coiled and partially folded state (Fig. 1). How do the authors determine and know each current state as partially folded, transient induction state, folded state, or bound conformational state? These signals could potentially be the results of a transition between multiple partially folded states or binding/unbinding to the device surface or SiNW. Additionally, what is the implication and role of the partially folded and pre-folding conformation? The authors need to elaborate on these.

Response: We thank this reviewer for the comments and suggestions. (i) It is an interesting question about the specific structure of the transient coiled and partially folded state. We think that the current states observed in Fig. 1 represent the conformation ensembles formed by the self-folding of apo-LC46. However, the distinguishment of each conformation ensemble as a specific structure requires instruments with higher time resolution. The energy landscape of apo-LC46 is relatively flat, where most of the local minimums were unable to cause significant effective gating voltage to be detected by our devices. In some cases, the conformation ensemble within the energy well of apo-LC46 may show a longer dwell time, which can be detected by our devices. (ii) The folding process and binding state can be distinguished by the comparison with control experiments as shown in **Fig. R18**. According to the Debye screening effect, the electrical double layer (EDL) formed in the ionic solution around the surface of silicon nanowires screens the charge signals from the solution environment beyond the Debye length, including random collisions of charged ions and charged molecules (*Nano Lett.* **2012**, *12*, 5245; *Nano Lett.* **2015**, *15*, 2143). Therefore, because of the lack of the strong interaction with SiNWs in this case, the charge interaction of the protein random collisions is screened by EDL and the conductance level of nanowires remains unchanged. In other words, the random collision of DQ47 or other molecules is averaged into the background noise as the other charged particles in the solution. (iii) The special current state observed in the Myc-Max and Myc-inhibitor experiments was actually an intermediate ensemble formed by the folding of LC46 between a free state and the fully folded binding state with DQ47. It is more appropriate to refer to this intermediate ensemble as encounter intermediate ensemble rather than pre-binding state. We have revised the descriptions in the revised manuscript.

Our Revision: We have added **Fig. R18** as **Supplementary Fig. 26** in the Supplementary Information. Relevant description has been added on **Page 11 in the main text**: *Based on these concentration-dependent signal patterns, we propose an interaction hypothesis (Fig. 1d) for LC46 and DQ47. At a very low DQ47 concentration (5 nM), LC46 may first undergo a loose interaction system and fold to a dominant Myc*, which is favourable for further specific binding. This encounter intermediate ensemble was further enlarged along with the increase of DQ47 concentration, according to the increase of Myc* population (5–100 nM DQ47 in Fig. 3c). As DQ47 concentration further increase, the equilibrium of the binding process would finally shift to the fully folded binding structure (100 nM – 5 μ M DQ47 in Fig 3c).*

9. (Page 9, lines 1-3). It is not obvious that 1* is a dynamic self-folding process? Why does the protein undergo a self-folding process at low and high temperatures? The authors need to provide a more convincing argument and experimental support for this.

Response: Thank this reviewer very much for the comments and suggestions. We apologize for our unclear explanations about state 1*. State 1* was a current state derived from the self-folding process, which occurred with the similar current level to state 1, but with different characteristic dwell time (as shown in Fig. 2c, two different peaks occurred in the dwell time distribution of the medium current level. the dominant one [dark green] was state 1, the other one was state 1*). After careful consideration about state 1*, we realized that the re-emerging state 1* at high temperatures might not be the same as that at lower temperature in spite of their similar current levels. As a result, we revised Figs. 2d and e as **Fig. R22**. State 1*_{LT} exhibited a similar current level to state 1, but with longer dwell times, and disappear at 35 °C.

Fig. R22 | Kinetic properties of the self-folding process of apo-LC46. **a**, Dwell time of different current states corresponding to different conformational intermediates ($n = 3$, number of the devices). **b**, Transition relationships between different conformational intermediate ensembles. Dotted lines indicate that the process may disappear at certain temperatures.

Our Revision: We have added **Fig. R22** as **Figs. 2d and e** in the main text. Relevant descriptions have also been revised on **Page 9** in the main text:

Two different populations (state 1 as well as state 1_{LT}^* at lower temperature or state 1_{HT}^* at higher temperature) were found in the dwell time distributions of the medium current state, indicating that at the same temperature, there were two conformational ensembles with similar effective gate voltage but different kinetic behaviours. The dynamic behaviour of state 1_{LT}^* and state 1_{HT}^* showed a distinct temperature-dependent nature. Interestingly, state 1_{LT}^* only existed at lower temperatures (25, 30 °C), and 1_{HT}^* emerged at higher temperature (45 °C), which were barely captured near physiological temperature (35–40 °C) (Figs. 2c and 2d).

Fig. 2 | Real-time monitoring of the c-Myc self-folding process. a, Real-time current

trajectory (time resolution of 17 μ s) of the c-Myc-modified device in a blank buffer solution at 25 $^{\circ}$ C (phosphate-buffered saline [PBS, $\times 0.01$, 8g/L NaCl, 0.2 g/L KCl, 1.44 g/L Na_2HPO_4 and 0.24 g/L KH_2PO_4], 5% dimethylsulfoxide [DMSO], pH = 7.4). The left panel shows a 60-s real-time trajectory, the middle panel shows a magnification of the clustered current signals, and the right panel is a histogram of the current data. **b**, Current pulses within the clustered signals at different temperatures. **c**, Dwell time distribution of the medium current states (States 1 [dark green], I_{LT}^* [light green] and I_{HT}^* [magenta]) at different temperatures. **d**, Dwell time of different current states corresponding to different conformational intermediates ($n = 3$, number of the devices). **e**, Transition relationships between different conformational intermediate ensembles. Dotted lines indicate that the process may disappear at certain temperatures.

10. The authors need to clarify what “n” represents in the Figures. Is it the number of devices, proteins, or signal segments?

Response: The “n” in the figures represents the number of devices, which is also shown in Supplementary Tables 1–6.

Our Revision: We have added the relevant descriptions about “n” to the legend of the figures in the main text as below.

Legend of Fig. 2d: Dwell time of different current states corresponding to different conformational intermediates ($n = 3$, number of the devices).

Legend of Fig. 3d: Population of the Myc–Max bound state at different DQ47 concentrations and different pHs ($n = 3$, number of the devices).

Legend of Fig. 3e: The activation energy of binding and dissociation were obtained by fitting the Arrhenius equation ($n = 3$, number of the devices).

Legend of Fig. 3f: The enthalpy change and entropy change of the dissociation process were obtained by linear fitting ($n = 3$, number of the devices).

Legend of Fig. 4d: Population of Myc-inhibitor binding states at different inhibitor concentrations ($n = 3$, number of the devices).

Legend of Fig. 5f: The population of Myc–Max binding states at different DQ47 concentrations in the presence of 100 μ M inhibitor ($n = 3$, number of the devices).

11. Similar to the previous point, the authors need to provide details on the charges and charge distribution of DQ47, and they get close to the SiNW when bound? How does binding result in substantial changes in the conductance of the devices?

Response: We thank this reviewer for the comments. The conformation of LC46 mainly

contains a random coil structure with short and transient helical structures as revealed by previous studies (*Sci. Rep.* **2016**, *6*, 22298 and *PLOS Comput. Biol.* **2013**, *9*, e1003249). After binding to DQ47 to form a stable heterodimer (as indicated by the complex structure: PDB: 1NKP), LC46 folds into helix-loop-helix conformation which dramatically shortens its radius of gyration. The charge distribution of the well folded LC46-DQ47 complex was depicted in the following figure since DQ47 only could not influence the current changes. Thus, we think that the substantial changes in the conductance were mainly affected by charge distribution changes induced by the dramatic conformational changes.

Our Revision: We have added relevant descriptions about the conformational change and conductance changes in the main text.

Pages 10-11 in the main text: *The gating voltage was mainly affected by charge distribution changes induced by the dramatic conformational changes.*

12. How do the authors know that DQ47 induces induction of LC46 without direct interaction? Need more data or convincing arguments.

Response: Thanks for the comment. We apologize for our inaccurate description about the induction which led to misunderstanding. We considered that DQ47 did interact with LC46 before specific binding. Because of the disordered nature of DQ47 and LC46, DQ47 did not initially form a tightly specific binding complex structure with LC46, but a probable loose interaction system. LC46 undergoes folding caused by the interaction with DQ47 to form the encounter intermediate ensemble, further forming a tight complex structure.

Our Revision: We have revised the description about the interaction DQ47 on **Page 11 in the main text:**

Based on these concentration-dependent signal patterns, we propose an interaction hypothesis (Fig. 1d) for LC46 and DQ47. At a very low DQ47 concentration (5 nM), LC46 may first undergo a loose interaction system and fold to a dominant Myc, which is favourable for further specific binding. This encounter intermediate ensemble was further enlarged along with the increase of DQ47 concentration, according to the increase of Myc* population (5–100 nM DQ47 in Fig. 3c). As DQ47 concentration further increase, the equilibrium of the binding process would finally shift to the fully folded binding structure (100 nM – 5 μ M DQ47 in Fig 3c).*

13. This manuscript shows multiple conformational populations of LC46, and thus it

could be feasible to determine these conformational states via crystal structure analysis. Are there any crystal structures of LC46 or LC46-DQ47 complexes available to support the findings in this manuscript?

Response: We thank this reviewer for the comments and suggestions. The CD spectra of Fig 2a in the literature (*Sci. Rep.* **2016**, *6*, 22298) showed that the apo state of c-Myc₃₇₀₋₄₀₉ (6 amino acids shorter than LC46) was disordered and the NMR data of Fig 5 in the literature (*Biochemistry* **2019**, *58*, 3144) also demonstrated the intrinsic disordered nature of free c-Myc₃₅₂₋₄₃₇. As a result, LC46 itself is disordered which precludes crystal structure determination. As LC46 and DQ47 can form stable complex, the corresponding crystal structure of Myc-Max heterodimer complex has been reported before (PDB: 1NKP).

14. The temperature dependence of intermediate states, binding, and dissociation constants could be further validated by FACS or fluoresces assaying experiments, which will provide stronger evidence.

Response: It is a good idea to use fluorescent experiments as demonstration for our results. We have tried very hard to carry out fluorescent experiments. Unfortunately, we were not successful in synthesizing the labeled LC46 due to low solubility. In addition, we are also worried that the attached fluorescent labels on the disordered peptides may affect their conformational properties. We will continue to try in future studies.

15. The LC46-inhibitor complexes show similar changes in current direction and amplitude (dI) as the LC46-DQ47 complex, indicating that both inhibitors have the same level of positive charges of DQ47. Why/how does the current drop for the LC46-inhibitor complex? What are the charges for 10074 and 1205 and how the charges are determined? To verify the signal direction and amplitude, the authors could repeat these experiments using synthetic compounds with additional positive or negative charges.

Response: We thank the reviewer for the comments and suggestions. In fact, neither 10074-A4 nor PKUMDL-YC-1205 have positive or negative charges. As DQ47 could induce the fully folded (more condensed) conformation of LC46, the similar changes in current dI indicated that these small molecules also have the potential to induce extended-to-condensed conformational changes of LC46. In the case of DQ47, LC46 folds into a more compact structure (fully folded complex). Coupled with additional positive charge carried by DQ47, the current level of LC46-DQ47 complex is smaller than that of LC46-inhibitor complex as shown in the competition experiments (Fig. 5

and Supplementary Figs. 35 and 36).

Our Revision: Relevant descriptions have been added in the main text as below.

Pages 10-11 in the main text: *The gating voltage was mainly affected by charge distribution changes induced by the dramatic conformational changes.*

Page 14 in the main text: *As the presence of DQ47 led to a fully folded (more condensed) conformation of LC46, this similar change pattern in current signals indicated that these small molecules also had the potential to cause extended-to-condensed conformational changes of LC46.*

Page 15 in the main text: *Since neither 10074-A4 nor PKUMDL-YC-1205 have total charge, this bistate-like signal with lower current level corresponded to the binding process between LC46 and DQ47 because of the more compact structure and the additional total positive charge carried with DQ47.*

16. (Figure 4, 5). In the presence of Max and inhibitors, the magnitude and levels of current signals were changed. i.e., current levels of LC46-Dq47 in Fig3a vs. Fig5, LC46-10074 in Fig4a vs Fig5, and LC46-1205 in Fig4b vs Fig5. How do the Max and inhibitor affect the conformational states of LD46? Why do current levels change for the same complex? (i.e., the magnitude of the bound current level for the LD46-DQ47/10074/1205 should be the same?).

Response: We thank the reviewer for the comments and suggestions. In comparison with absolute values of the current levels, we focus more on the relative relationship between different current states. Because the absolute value of currents will deviate with different devices and the testing environments, the relative value can better reflect the relationship between different species in the same testing system. In this work, in contrast to the LC46-inhibitor complexes, the LC46-DQ47 complex can form a more condensed and stable helical structure, making the global positive charge denser on the surface of the SiNW and leading to the formation of a lower conductivity state. The concentration dependence also indicates that, at high concentrations of DQ47, the signal changed from a bistable state with high conductivity to that with low conductivity.

17. Since this approach allows long-term monitoring of protein fluctuations, it would be nice to see the dynamic disorder of protein fluctuations (e.g., dwell times vs. time, color-coded enzyme behaviors vs time, [dx.doi.org/10.1021/ja311604j](https://doi.org/10.1021/ja311604j), J. Am. Chem. Soc. 2013, 135, 7861–7868, [dx.doi.org/10.1021/ja211540z](https://doi.org/10.1021/ja211540z), J. Am. Chem. Soc. 2012, 134, 2032–2035).

Response: Thanks a lot for the constructive comment about the dynamic disorder. We have plotted the dwell time vs. time properties of the binding process from the encounter intermediate ensemble to the fully folded binding complex form by LC46 and DQ47 (**Fig. R23**). The averaged dwell times $\langle t \rangle$ were calculated from 1-second data. The figure illustrated the dynamic disorder nature of single-molecule events. $\langle t_{\text{econ}} \rangle$ showed a wider fluctuation compared with $\langle t_{\text{bind}} \rangle$, indicating the ensemble properties of encounter intermediates.

Fig. R23 | Variation in the dwell-time mean values $\langle t \rangle$ of DQ47-LC46 binding kinetics. These values were arithmetic mean values calculated from dwell times within indicated 1-s data over a 50-second interval at 37 °C with 1 μM DQ47. The corresponding τ derived from the exponential fit of each state is indicated by dashed lines (red for binding state and blue for encounter state through dissociation).

Our Revision: We have added the **Fig. R23** as **Supplementary Fig. 28** in the Supplementary Information. Relevant descriptions have been added in the main text as below.

Page 11 in the main text: *As the arithmetic averaged dwell times were calculated second by second, a dynamic disorder analysis was performed to explore the fluctuation of the conformational changes in the binding equilibrium⁵⁸. The arithmetic averaged dwell times of Myc* exhibited a distribution with a relatively wide range, implying a larger conformational fluctuation of the encounter intermediate ensemble in comparison with the binding complex (Supplementary Fig. 28).*

[58] Sims, P. C. *et al.* Electronic measurements of single-molecule catalysis by cAMP-dependent protein kinase A. *J. Am. Chem. Soc.* **135**, 7861-7868 (2013).

We would like to take this opportunity to thank this reviewer very much for all the time involved and important suggestions to improve the manuscript. We hope this reviewer will find this revised version satisfactory.

Sincerely,

The Authors

-----End of Response to Reviewer #4-----

REVIEWER COMMENTS

Reviewer #1 (Remarks to the Author):

I am happy with the extensive revisions of the manuscript. I recommend publication in Nature Communications.

Reviewer #2 (Remarks to the Author):

The authors have adequately addressed my suggestions in the revised version of the manuscript.

Reviewer #3 (Remarks to the Author):

This thorough review addresses issues I described earlier. I'm now convinced, and hope the authors include the details requested by the reviewers in their final version.

Reviewer #4 (Remarks to the Author):

The authors have addressed most of my concerns satisfactorily. However, there are a few remaining points that need to be addressed in order to fully convince readers.

(Q6) In the initial manuscript, all experiments were described as being performed in PBS. However, it has been changed to 0.01XPBS without any explanation. Is this a simple error? It is important to determine whether the protein exhibits native activity at 0.01XPBS, which is a non-physiological salt concentration, within the temperature range of 25-40°C. To ensure the preservation of protein activity, the authors should conduct a standard biological activity assay using various PBS concentrations and temperatures.

(Q7, Q11, Q15, Q16) The manuscript presents electrical signals that are dynamic and consistently repeating with the same amplitude, showing very narrow and distinguishable two or three current distributions. This raises the question of why the dynamics of protein conformational changes and protein-partner interactions appear to be highly consistent. Shouldn't there be random 3D motions and rational motions during the interactions, leading to a larger variation in signal amplitude and merged, indistinguishable signal distributions?

Additionally, the authors have not provided an answer to the question of what changes, how many charges are involved, and what dynamic motions those charges undergo relative to the SiNW. Instead, the authors repeatedly claim that the signals and their changes originate from an "unknown" charge distribution resulting from "unknown" global conformational changes. The authors need to elaborate and provide more explanations regarding this issue. It would be possible to (1) calculate the protein's charge residues and total surface charges, (2) estimate their motions and distances from the SiNW during the protein's conformation, and (3) establish a connection between conformational changes of the protein or protein-partner complex and signal generation. See the following references: *J. Phys. Chem. B* 2010, 114, 9, 3330–3333, *Nano Lett.* 2013, 13, 2, 625–631.

Point-to-point response to the reviewers' comments (in blue)

We thank the four reviewers for their constructive comments and suggestions. We have revised our manuscript accordingly. Here is a brief summary of the major changes in the revised manuscript followed by the point-to-point response.

1. We have made revisions of the descriptions about the current state as suggested by reviewer #4. Please see Pages 7 and 17 in the revised main text.
2. We have added the comparison of the binding affinity under different PBS concentrations. Please see Page 7 in the revised main text and Page 39 in the revised Supplementary Information.
3. We have added descriptions about the total charge of the peptides in the revised manuscript. Please see Pages 8 and 11 in the revised main text and Page 4 in the revised Supplementary Information.
4. We have added the Methods section in the main text. Please see Pages 25–27 in the revised main text.

Point-to-point response to the reviewers' comments (in blue)

REVIEWER COMMENTS

Reviewer #1 (Remarks to the Author):

I am happy with the extensive revisions of the manuscript. I recommend publication in Nature Communications.

Response: We thank the reviewer for the positive comments.

Reviewer #2 (Remarks to the Author):

The authors have adequately addressed my suggestions in the revised version of the manuscript.

Response: We thank the reviewer for the positive comments.

Reviewer #3 (Remarks to the Author):

This thorough review addresses issues I described earlier. I'm now convinced, and hope the authors include the details requested by the reviewers in their final version.

Response: We thank the reviewer for the positive comments. We have included the details requested by the reviewers in our latest version.

Reviewer #4 (Remarks to the Author):

The authors have addressed most of my concerns satisfactorily. However, there are a few remaining points that need to be addressed in order to fully convince readers.

(Q6) In the initial manuscript, all experiments were described as being performed in PBS. However, it has been changed to 0.01X PBS without any explanation. Is this a simple error? It is important to determine whether the protein exhibits native activity at 0.01XPBS, which is a non-physiological salt concentration, within the temperature range of 25-40°C. To ensure the preservation of protein activity, the authors should conduct a standard biological activity assay using various PBS concentrations and temperatures.

Response: We thank the reviewer for the suggestions about the PBS concentration. In order to fully detect the conformational change of LC46 and its interaction with other ligands, experiments were conducted with various concentrations of phosphate-buffered saline (PBS) to investigate the influence of ionic strength and provide an optimal signal-noise ratio. The experimental results are illustrated in Supplementary Fig. 20, which is also given below, showing an optimal signal-noise ratio in 0.01× PBS with a Debye length of 7.5 nm. The signals were not obvious under high salt concentrations (Supplementary Fig. 20). We agree that it is important to determine whether the protein exhibits the binding affinity in 0.01× PBS. We carried out SPR experiments under 0.01× PBS (0.0× PBS, 0.2% T20, 5% DMSO, pH7.4) and the kinetic analysis results are shown in **Table R1**. The K_D of Myc and Max peptides under 0.01× PBS is about 347 nM, which is close to that under 1 × PBS (351 nM).

Supplementary Fig. 20 | PBS concentration-dependent experiments. a–c, Real-time current trajectories of a single-Myc modified device in different PBS solutions (PBS buffer [8g/L NaCl, 0.2 g/L KCl, 1.44 g/L Na₂HPO₄ and 0.24 g/L KH₂PO₄], pH = 7.4, 5% DMSO): 1× PBS (a), 0.1× PBS (b), 0.01× PBS (c). The right column is the 60-s current data, and the left column is the magnified view of the selected area.

Table R1. Binding affinity of Myc and Max at different PBS concentrations.

PBS pH 7.4 (5% DMSO)	K_D (nM)
×1	351 ± 28
×0.01*	374 ± 147

*0.01×PBS, 0.2% T20, 5% DMSO, pH7.4

Our Revision: Table R1 has been added as Supplementary Table 2 in the Supplementary Information. Relevant descriptions have been added in the main text.

Page 7 in the main text: *Kinetic analysis results from surface plasmon resonance (SPR) experiments indicated that the binding activity of Myc and Max remained under this low salt concentration (Supplementary Table 2).*

(Q7, Q11, Q15, Q16) The manuscript presents electrical signals that are dynamic and consistently repeating with the same amplitude, showing very narrow and distinguishable two or three current distributions. This raises the question of why the dynamics of protein conformational changes and protein-partner interactions appear to be highly consistent. Shouldn't there be random 3D motions and rational motions during the interactions, leading to a larger variation in signal amplitude and merged, indistinguishable signal distributions?

Response : We thank the reviewer for the valuable comments. We agree that the disordered proteins can adopt numerous indistinguishable 3D conformations. The folding signal emerged with short durations during the motion of Myc in the blank buffer solution (Supplementary Fig. 19). In most periods, we observed a single conductivity state. We consider that this single conductivity state may not correspond to a specific conformation but a conformational ensemble with multiple conformations. During the single current state, the disordered protein lied within an energy landscape without significant energy wells, and the transitions between conformations might be too fast (on the sub- μ s to μ s timescale) for us to capture.

The oscillatory signal aroused when the disordered protein transiently occupies a larger potential energy well on the energy landscape, resulting in a distinct conformational ensemble with a longer dwell time. We considered that there might be various sub-ensembles within the conformational ensemble, but the transition frequency between these sub-ensembles is faster than the sampling rate of our methods, requiring instruments with higher time resolution. We have also observed different conformational ensembles converging to the same current level, indicating distinct dynamic behaviors (as shown in Supplementary Fig. 24).

Furthermore, when interacting with partner molecules or ligands, specific conformational ensembles can be stabilized or induced by these molecules. These oscillation states represented several conformational ensembles rather than individual conformations. We considered that these conformational ensembles are different from the former conformational ensembles observed in the blank buffer.

Our Revision: We have added relevant revisions in the manuscript as below.

Page 7 in the main text: *In the measurement of Myc-modified devices, the current trajectories showed a long-last single current stage with transient clustered signals (in $0.01 \times$ PBS, 5% DMSO, pH = 7.4) (Fig. 1c, Fig. 2a, and Supplementary Fig. 19c).*

Page 7 in the main text: *The prolonged single current state observed was not considered to be a specific conformation but rather a conformational ensemble encompassing multiple conformations. In the period of the single current state, the LC46 peptide resided within an energy landscape without prominent energy wells. The transitions between different conformations occurred at a rapid pace, on the timescale ranging from sub- μ s to μ s. These conformational changes were beyond the limitation of the sampling rate and difficult for us to capture. The cluster oscillatory signals were considered to arise when the peptide briefly occupies the deeper potential wells on the energy landscape, leading to the formation of distinct conformational ensembles characterized by longer dwell times, which can be captured.*

Page 17 in the main text: *The conformational ensembles with different dynamic behaviours were achieved to show the reconstruction of the energy landscape of the IDR.*

Additionally, the authors have not provided an answer to the question of what changes, how many charges are involved, and what dynamic motions those charges undergo relative to the SiNW. Instead, the authors repeatedly claim that the signals and their changes originate from an "unknown" charge distribution resulting from "unknown" global conformational changes. The authors need to elaborate and provide more explanations regarding this issue. It would be possible to (1) calculate the protein's charge residues and total surface charges, (2) estimate their motions and distances from the SiNW during the protein's conformation, and (3) establish a connection between conformational changes of the protein or protein-partner complex and signal generation. See the following references: *J. Phys. Chem. B* 2010, 114, 9, 3330–3333, *Nano Lett.* 2013, 13, 2, 625–631.

Response: We thank the reviewer for the valuable comments. We agree that (1) Based on the charges of amino acid residues under physiological pH conditions, both the IDR LC46 and DQ47 employed in our study possess a net charge of +2e. However, as they do not exhibit stable or compact 3D structures in solution, it is difficult to accurately calculate surface charges. (2) According to the references (*J. Phys. Chem. B* 2010, 114, 9, 3330, *Nano Lett.* 2013, 13, 2, 625.), we attempted to calculate the influence of selected structures on the C-terminus (linkage end) based on the Coulombic electrostatic potential model (**Eq. R1**). The conformations employed in our calculations are simulated based on the NMR results of LC46 (conformations generated from

<https://csrosetta.bmr.bio/submit>) as shown in **Fig. R2**, where the purple conformation represents a partially-folded intermediate (apo state), and the green one represents the conformation in the Myc-Max complex. The results are shown in **Table R2**. The Myc-Max complex exhibits a higher Coulombic potential, occupying a lower conductive state in our study, suggesting that the Coulombic potential can provide some reference information. However, it should be noted that the peptides employed are smaller in size in comparison with SiNWs (as shown in Supplementary Fig. 18) and the sensing part of SiNWs cannot be approximated to a single point, which results in a much more complex condition between peptides and SiNWs. Coulombic potential method based on point charge model might be inadequate in this case. A more accurate theoretical model and rational designed experiments will be constructed in our further study. (3) Currently it is impossible for us to run molecular dynamics simulations to a time-scale that is comparable to the experimental time-scale due to huge computational time required. Thus, conformations from simulations and single-molecule electrical experiments may not be directly compared. Furthermore, the current states captured in the experiments represented conformational ensembles of IDR rather than specific conformations. Therefore, establishing a correlation between conformational changes and conductive state transitions necessitate further theoretical and experimental investigations.

$$\Delta\Phi \cong \frac{1}{4\pi\epsilon} \frac{q}{r} e^{-r/\lambda_D} \quad (\text{Eq. R1})$$

Fig. R2 | The conformations used in the Coulombic potential calculations. Purple conformation represents a partially-folded structure in apo state (generated from <https://csrosetta.bmr.bio/submit>). Green and Blue conformations represent LC46 and DQ47 in the Myc-Max complex (PDB 1NKP).

Table R2. The electrostatic potential at C-terminal of the peptides in 0.01× PBS.

$\Delta\Phi$	Full calculation (mV)
Apo state	-11.6
Binding complex	-13.7
Difference	+1.9

Our Revision: The relevant descriptions about the total charge of the peptides have been added in the revised manuscript.

Page 8 in the main text: *LC46 (pI = 8.94) used in these experiments was positively charged (+2e) in PBS at pH = 7.4.*

Page 11 in the main text: *Because of the global positive charge of DQ47 (+2e at pH 7.4)*

Page 4 in the Supplementary Information: *LC46 and DQ47 both carry +2e at pH 7.4.*

Finally, we would like to thank all the referees very much for their patience, precious time and kind support.

REVIEWERS' COMMENTS

Reviewer #4 (Remarks to the Author):

The authors have addressed my comments and revised the text properly.